# Extending Foundation Models to Low-Resource Languages: Vocabulary Expansion and Policy Optimization

## Abstract

Large language models have achieved remarkable success in multilingual machine translation, yet they encounter persistent difficulties in low-resource languages, especially those with unique scripts and complex morphology, arising from the lack of subword-level segmentation. We propose Multilingual Translation Policy Optimization (MtPO), a comprehensive three-stage framework: continued pre-training grows low-resource vocabularies, improving compression and inference efficiency; curriculum SFT raises task difficulty while preserving both general and specialized translation skills; and RL optimization counters length bias and diversity collapse in GRPO, reinforced with RLVR. The RL component supplements semantic rewards with fast deterministic constraints on length ratio, structural token retention (HTML/Markdown), target-language validity, and code-mixing to harden models against messy real-world prompts. MtPO couples entropy-tempered advantages, temporal decay, asymmetric clipping, and token-wise reward normalization to sustain early exploration before settling, while RLVR enforces reliable outputs without harming translation quality. Experiments confirm notable gains in tokenization efficiency, translation quality, and exploration–exploitation balance, marking a substantive step forward for multilingual models serving underrepresented languages and practical deployments.

## 1 Introduction

Neural machine translation has substantially advanced through the integration of large language models (LLMs). However, their real-world deployment remains constrained by three fundamental limitations: insufficient support for low-resource languages, disproportionately high computational costs, and fragile performance in generating outputs for structured or mixed-language content.

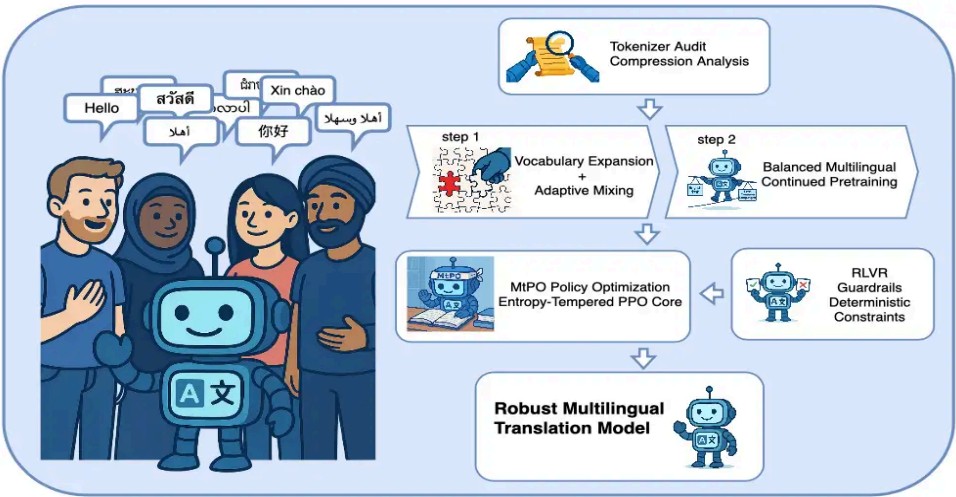

Current multilingual systems, including state-of-the-art models like GPT-4 (OpenAI et al., 2024), PaLM 2 (Anil et al., 2023), DeepSeek (DeepSeek-AI et al., 2025), and Qwen2.5 (Yang et al., 2024), demonstrate impressive capabilities in high-resource languages but exhibit significant performance degradation on low-resource languages. These models use tokenizers and pre-training corpora optimized for high-resource scripts, leaving languages with distinct writing systems and morphological structures systematically underserved. For languages such as Khmer, Myanmar (Burmese), and Thai, this mismatch manifests as dramatically longer token sequences, increased inference latency, and unstable translation quality due to insufficient representation in both vocabulary and training data.

Prior work has tackled these challenges from two directions. Instruction-tuned chat models pursue breadth: open releases such as Apertus, Tower-Plus, and Aya emphasize transparency and multi-domain utility, while commercial systems expose translation via general-purpose conversational interfaces. In parallel, specialized MT stacks, such as Seed-X (Cheng et al., 2025a), Qwen-MT, Hunyuan-MT (Zheng et al., 2025), and region-focused efforts like COMPASS-V2, SEA-LION, Sailor2 (Dou et al., 2025a), and SeaLLMs, curate domain data, bespoke decoding, or bilingual reinforcement learning to maximize fidelity. These approaches advance the state of multilingual translation, yet each falls short in production settings, where reliance on fixed translation-style instructions results in overly narrow and inflexible use cases.

**Core challenges and limitations.** Despite significant advances in multilingual NLP, three fundamental issues continue to limit the effectiveness of current approaches for low-resource languages:

**(1) Tokenization inefficiency:** Existing tokenizers, designed primarily for English and other high-resource languages, perform poorly on morphologically rich and non-segmented scripts. For instance, a typical Khmer sentence requires 3 to 4 times more tokens than its English equivalent when processed by standard multilingual tokenizers. This inefficiency directly translates to increased computational costs, longer inference times, and memory bottlenecks that lead to significant resource wastage in constrained environments.

**(2) Training data imbalance:** While general-purpose LLMs achieve broad language coverage, their pre-training distributions remain heavily skewed toward English and other high-resource languages. This imbalance persists even after supervised fine-tuning, resulting in models that struggle with low-resource languages' unique linguistic phenomena, cultural contexts, and domain-specific terminology.

**(3) Reinforcement learning challenges:** When applying reinforcement learning to improve translation quality, existing methods often suffer from entropy collapse and length bias. Models trained with standard RL objectives tend to generate overly verbose or repetitive outputs, while failing to maintain the structural integrity required for real-world applications (such as preserving HTML markup or maintaining appropriate language targeting). Related RL-based translation efforts include ExTrans (Wang et al., 2025).

To resolve these bottlenecks, we introduce **Multilingual Translation Policy Optimization (MtPO)**, a unified pipeline that couples tokenizer expansion, balanced continued pre-training, and entropy-aware reinforcement learning. MtPO is complemented by **Reinforcement Learning with Verifiable Rewards (RLVR)**, a guardrail suite that applies deterministic length, format, language-identification, and code-mixing checks during optimization. Techniques such as multilingual test-time scaling (Bajpai & Chakraborty, 2025) address decoding-time improvements but do not resolve training-time structural failures. The resulting system preserves the breadth of instruction-tuned LLMs while achieving translation robustness comparable to specialized MT services.

**Contributions.** Our work delivers three technical advances: **(1)** a systematic tokenizer audit and expansion strategy that substantially improves decoding efficiency across eight low-resource languages, while maintaining robust coverage for English; **(2)** a reinforcement learning objective that integrates temperature consistency, asymmetric clipping, and entropy-shaped credit assignment to avert exploration collapse during translation-oriented reinforcement learning; and **(3)** an RLVR-based filtering mechanism that enforces structural constraints while simultaneously maintaining high-quality semantic rewards.

MtPO's holistic design yields strong empirical gains. In Table 3, LLM-7B-MtPO achieves state-of-the-art average BLEU scores across 90 FLORES-Plus directions while preserving competitive zero-shot accuracy on reasoning benchmarks such as BBH, CMMLU, and HellaSwag. These results

demonstrate that principled tokenizer augmentation and entropy-aware policy optimization can close the gap between instruction-tuned LLMs and production-grade multilingual MT.

## 2 RELATED WORK

Recent multilingual LLMs aim to strengthen translation alongside general instruction following, either by dedicated specialization or by carefully combining both objectives. *Apertus* emphasizes openness, transparency, and broad multilingual coverage—including many lower-resource languages—rather than introducing explicit translation-specific prompt templates or "hooks" (Team, 2025). *TOWER+* extends TOWER to balance translation specialization with general-purpose capabilities via a staged recipe comprising continued pretraining, supervised fine-tuning, preference optimization, and reinforcement learning with verifiable rewards(Rei et al., 2025). Similarly, *EMMA-500* and *LLaMAX* demonstrate effective approaches to massively multilingual adaptation(Ji et al., 2025; Lu et al., 2024), while *Aya* provides instruction fine-tuned multilingual capabilities(Üstün et al., 2024).

**Regional Models for ASEAN Languages.** Several initiatives specifically target Southeast Asian languages with curated data and alignment efforts. *COMPASS-V2* focuses on SEA languages (and e-commerce scenarios), combining curated corpora, tokenizer, and architecture choices to sustain quality across diverse tasks (Maria, 2025). *SEA-LION* (Ng et al., 2025), *Sailor2* (Dou et al., 2025b), and *SeaLLMs 3* (Zhang et al., 2024) broaden SEA coverage and cultural alignment through continual pretraining, staged instruction tuning, and region-aware sampling. *Babel* (Zhao et al., 2025), a broad multilingual LLM including SEA languages, shows how expanded coverage and efficient scaling can complement regional efforts.

**Dedicated Machine Translation Systems.** Pure machine translation models push quality forward with specialized architectures that prioritize bilingual fidelity. Seed-X (Cheng et al., 2025b), Qwen-MT (Qwen, 2024) and Hunyuan-MT (Hunyuan, 2024) show that translation-centric designs can rival commercial systems. Recent compact models such as X-ALMA (Xu et al., 2025) further improve efficiency through modular adapters and selective rejection mechanisms, highlighting optimization levers that remain effective at smaller scales. These systems establish strong translation baselines and reveal architectural ingredients that can be ported to more general models.

**Limitations vs. MtPO.** Earlier "traditional" MT pipelines—phrase-based SMT systems exemplified by Moses (Koehn et al., 2007) and early neural MT architectures such as GNMT or bilingual Transformer models (Wu et al., 2016; Vaswani et al., 2017)—provide high BLEU on bilingual corpora but expose three disadvantages relative to MtPO. (1) Their vocabularies are fixed to word alignments or SentencePiece models trained on Euro-centric data, so segmentation for ASEAN morphologies remains brittle, limiting compression benefits when expanding to agglutinative or code-mixed inputs. (2) Their objectives target translation fidelity alone; dialog grounding, instruction following, or formatting guarantees must be handled by downstream components, which makes them unreliable when users interleave translation with general-purpose reasoning. (3) Reinforcement learning or minimum-risk training in these systems optimizes metric surrogates (BLEU, TER) without verifiable constraints, leaving practical failure modes—overlong responses, markup corruption, language-ID drift—unaddressed at decode time. In contrast, MtPO's tokenizer-aware continued pretraining, unified SFT+RL stack, and RLVR constraints allow a single policy to cover translation, instruction following, and safety-critical formatting in low-resource languages.

**RL for Machine Translation.** Early reinforcement learning adaptations such as minimum risk training directly optimized sequence-level metrics to narrow the gap between training and evaluation (Shen et al., 2016). More recent work revisits preference-driven optimization for translation: contrastive preference optimization sharpens reward shaping for machine translation outputs (Xu et al., 2024a), RLHF pipelines are tailored for cost-sensitive translation preference modeling (Xu et al., 2024b), and preference-driven alignment strategies further refine translation quality (Zhu et al., 2024). Diagnostic studies interrogate the weaknesses of RL-based objectives and evaluation protocols (Wu et al., 2018), while multilingual preference optimization extends these ideas across languages (Dang et al., 2024).

# 3 METHODS

## PIPELINE OVERVIEW

We integrate continued pre-training, entropy-aware reinforcement learning, and verifiable constraints into a unified MtPO pipeline: expand the tokenizer, refresh the model with additional pre-training, and finish with SFT plus RLVR to support downstream alignment.

## TOKENIZER-AWARE CONTINUED PRETRAINING

Continued pre-training adapts large language models to specialized domains while retaining broad linguistic competence(Conneau et al., 2020a). For low-resource settings, vocabulary design remains decisive: introducing language-specific tokens better captures morphological structure(Conneau et al., 2020b; Fan et al., 2020; Devlin et al., 2019), directly addressing tokenization inefficiency through improved subword segmentation(Sennrich et al., 2016; Kudo & Richardson, 2018). **Stage 1 (Vocabulary Expansion).** We expand the Qwen2.5-7B tokenizer(Yang et al., 2024) by adding language-specific tokens for eight low-resource languages, creating the Qwen2.5-7B-8Langs tokenizer, following established multilingual practices(Team, 2022). **Stage 2 (Balanced Training).** A 1:1 balance between English and low-resource corpora preserves English proficiency while enhancing support for low-resource languages. The expanded tokenizer adds 3k–4k tokens per targeted language, achieving $2.1\times$–$5.4\times$ compression improvements Table 1, which translate to reduced inference latency and improved throughput.

## POST-TRAINING OPTIMIZATION WITH RLVR

**Supervised Fine-Tuning.** Post-training combines supervised fine-tuning and reinforcement learning without additional submodules. Our SFT stage trains on a 7M-sample mixture (5:1 general instructions vs. multilingual data) under a three-phase curriculum—**(1) basic instruction following**, **(2) ASEAN language translation**, and **(3) merged translation+instruction tasks**—with sampling gradually shifting from simple to complex. The reward model captures both translation quality and instruction-following capability, following RLHF best practices(Ouyang et al., 2022; Ziegler et al., 2019), and uses ten preference categories spanning accuracy, fluency, terminology, format, and code-mixing (Appendix A.3).

**RLVR Motivation and Components.** Deployment logs reveal four dominant failure modes: (i) translations that exceed length budgets, (ii) markup corruption on HTML/Markdown inputs, (iii) drifting from the target language or heavy code mixing, and (iv) reward hacking where semantic quality drops despite good BLEU proxies. MtPO therefore wraps reinforcement learning in a verifiable reward layer. Reinforcement Learning with Verifiable Rewards (RLVR) augments the semantic reward $R_{\mathrm{mt}}$ with deterministic verifiers $r_k$ that are cheap to evaluate and influence optimization through sample selection:

$$\begin{aligned}
R_{\mathrm{RLVR}}(x, y) = {} & R_{\mathrm{mt}}(x, y) \\
& + \lambda_{\mathrm{len}}\, r_{\mathrm{len}}(x, y) + \lambda_{\mathrm{fmt}}\, r_{\mathrm{fmt}}(x, y) \\
& + \lambda_{\mathrm{lid}}\, r_{\mathrm{lid}}(y) + \lambda_{\mathrm{mix}}\, r_{\mathrm{mix}}(y).
\end{aligned} \tag{1}$$

where $x$ and $y$ denote the source and target translations, $R_{\mathrm{mt}}(x, y)$ is the semantic translation reward from the preference model, and $\lambda_{\mathrm{len}}, \lambda_{\mathrm{fmt}}, \lambda_{\mathrm{lid}}, \lambda_{\mathrm{mix}}$ are weighting coefficients for each verifier. Each verifier targets one failure mode: (i) $r_{\mathrm{len}}$ enforces $\rho = |y|/|x| \in [\alpha, \beta]$, (ii) $r_{\mathrm{fmt}}$ preserves HTML/Markdown markers, (iii) $r_{\mathrm{lid}}$ runs a lightweight language-ID model to confirm $\ell_t$ with confidence above $\theta_{\mathrm{lid}}$, and (iv) $r_{\mathrm{mix}}$ penalizes intra-sentence code mixing. Detailed formulas appear in Appendix A.2. All verifiers operate on strings, so they remain model-agnostic and run as batched regex/LID checks. For each prompt we sample $K$ candidates, score with $R_{\mathrm{RLVR}}$, and keep the top $G$ diverse hypotheses to preserve exploration while discarding violations before optimization.

**Alignment Objective.** The corresponding objective combines the clipped surrogate with entropy regularization:

$$\mathcal{J}_{\text{MtPO}}(\theta, \tau) = \mathbb{E}_{(q,a)\sim\mathcal{D},\ \{o_i\}_{i=1}^{G}\sim\pi_{\theta_{\text{old}}}^{\tau}}\left[\frac{1}{\sum_{i=1}^{G}|o_i|}\sum_{i=1}^{G}\sum_{t=1}^{|o_i|}\right.$$

$$\min\left(r_{i,t}^{\tau}(\theta)\,\hat{A}_{i,t}^{(\text{micro}+H)},\ \text{clip}(r_{i,t}^{\tau}(\theta), 1-\epsilon_{\text{low}}, 1+\epsilon_{\text{high}})\,\hat{A}_{i,t}^{(\text{micro}+H)}\right)\Bigg]$$

$$+\ \beta\,\mathbb{E}_{(q,a),\ \{o_i\}\sim\pi_{\theta_{\text{old}}}^{\tau}}\left[\frac{1}{\sum_{i=1}^{G}|o_i|}\sum_{i=1}^{G}\sum_{t=1}^{|o_i|}H(\pi_\theta(\cdot\mid q, o_{i,<t}))\right] \tag{2}$$

where $\theta$ denotes the policy parameters, $\tau$ is the temperature for sampling, $\mathcal{D}$ is the training data distribution, $(q,a)$ are query-answer pairs, $\{o_i\}_{i=1}^{G}$ are $G$ candidate outputs sampled from the old policy $\pi_{\theta_{\text{old}}}^{\tau}$, $|o_i|$ denotes the length of candidate $i$, $t$ indexes token positions, $\epsilon_{\text{low}}$ and $\epsilon_{\text{high}}$ are clipping bounds, and $\beta$ is the entropy regularization coefficient. The objective uses

$$r_{i,t}^{\tau}(\theta) = \frac{\pi_\theta^\tau(o_{i,t}\mid q, o_{i,<t})}{\pi_{\theta_{\text{old}}}^\tau(o_{i,t}\mid q, o_{i,<t})},$$

$$\hat{A}_{i,t}^{(\text{micro}+H)} = \frac{R_{i,t} - \text{mean}_{\text{group}}(\{R_{i,t}\}_{i=1}^{G})}{\text{std}_{\text{microbatch}}(\{R_{i,t}\}_{i=1}^{G}) + \epsilon}\cdot\left(1 + \alpha\cdot H(\pi_\theta(o_{i,t}\mid q, o_{i,<t}))\cdot\gamma^t\right).$$

Here $r_{i,t}^{\tau}(\theta)$ is the importance ratio comparing the current policy $\pi_\theta^\tau$ to the old policy $\pi_{\theta_{\text{old}}}^\tau$ at token $o_{i,t}$, $R_{i,t}$ denotes the token-level reward from the preference model(Liu et al., 2024), $\hat{A}_{i,t}^{(\text{micro}+H)}$ is the entropy-modulated advantage estimator, $\alpha$ is the entropy scaling coefficient, $\gamma$ is the entropy decay factor, $\epsilon$ is a small constant for numerical stability, and $H(\pi_\theta(o_{i,t}\mid q, o_{i,<t}))$ is the entropy of the policy at position $t$. The advantage estimator (defined above) normalizes by group mean and microbatch standard deviation, ensuring heterogeneous prompts share comparable gradient scales. The entropy multiplier $(1 + \alpha H\gamma^t)$ keeps early tokens exploratory before gradually tightening. The same temperature $\tau$ is used in both sampling and importance ratios, preserving the martingale property required for stable critic-free RL. Combined with global length normalization, these changes remove the verbosity bias typical of GRPO objectives. MtPO introduces three architectural choices: (1) **asymmetric clipping with layered entropy shaping**, maintaining high entropy at the beginning of decoding but allowing sharper updates later; (2) **microbatch-level normalization**, stabilizing advantages across diverse prompts and keeping the surrogate length-invariant; and (3) **constraint-aware data filtering through RLVR**, replacing DAPO's hand-crafted length penalties with verifiable filters so the optimizer never trains on samples violating production policies. Detailed derivations appear in Appendix A.4, and ablations in Section 4 confirm each component's contribution.

**Optimization Recipe.** Algorithm 1 summarizes one MtPO iteration: it collects trajectories with temperature-controlled sampling, computes entropy-aware advantages and temperature-consistent ratios, then aggregates losses with global length normalization plus an entropy floor.

---

**Algorithm 1** Multilingual Translation Policy Optimization (MtPO)

---

**Require:** Initial policy parameters $\theta_0$, temperature $\tau$, entropy coefficient $\beta$, entropy decay $\gamma$, entropy scaling $\alpha$, clipping parameters $\epsilon_{\text{low}}, \epsilon_{\text{high}}$
1: Initialize $\theta \leftarrow \theta_0$
2: **while** not converged **do**
3:     Sample translation prompts $(q,a) \sim \mathcal{D}$ and set $\theta_{\text{old}} \leftarrow \theta$
4:     **for** each $q$ **do**
5:         Sample $G$ candidates $\{o_i\}_{i=1}^{G} \sim \pi_{\theta_{\text{old}}}^{\tau}(\cdot|q)$ using temperature $\tau$
6:         Compute token rewards $R_{i,t}$ with the reward model
7:         Normalize rewards: $\mu_{\text{group}} = \text{mean}(\{R_{i,t}\}_{i=1}^{G})$, $\sigma_{\text{microbatch}} = \text{std}(\{R_{i,t}\}_{i=1}^{G})$
8:         **for** each $o_i$ and position $t$ **do**
9:             $H_{i,t} \leftarrow H(\pi_\theta(o_{i,t}\mid q, o_{i,<t}))$
10:            $\hat{A}_{i,t} \leftarrow \frac{R_{i,t}-\mu_{\text{group}}}{\sigma_{\text{microbatch}}+\epsilon}(1+\alpha H_{i,t}\gamma^t)$
11:            $r_{i,t}^{\tau} \leftarrow \frac{\pi_\theta^\tau(o_{i,t}|q, o_{i,<t})}{\pi_{\theta_{\text{old}}}^\tau(o_{i,t}|q, o_{i,<t})}$
12:            $L_{i,t}^{\text{clip}} \leftarrow \min\left(r_{i,t}^{\tau}\hat{A}_{i,t}, \text{clip}(r_{i,t}^{\tau}, 1-\epsilon_{\text{low}}, 1+\epsilon_{\text{high}})\hat{A}_{i,t}\right)$
13:        **end for**
14:    **end for**
15:    $N \leftarrow \sum_{i=1}^{G}|o_i|$
16:    $\mathcal{L}_{\text{MtPO}} \leftarrow -\frac{1}{N}\sum_{i,t}L_{i,t}^{\text{clip}} - \beta\frac{1}{N}\sum_{i,t}H(\pi_\theta(\cdot\mid q, o_{i,<t}))$
17:    Update $\theta \leftarrow \theta - \eta\nabla_\theta\mathcal{L}_{\text{MtPO}}$
18: **end while**
19: **return** $\theta$

---

**Alignment Theoretical Properties.** We now analyze MtPO through the lens of policy-optimization theory (Schulman et al., 2017) and multilingual modeling dynamics (Conneau et al., 2020b), focusing on (i) the geometry of the surrogate gradient, (ii) the integration of deterministic verifiers, and (iii) the effect of entropy shaping on tokenizer efficiency.

**Gradient geometry.** The surrogate gradient adopted by MtPO is

$$\nabla_\theta \mathcal{J}^{\mathrm{surr}}_{\mathrm{MtPO}} = \mathbb{E}\left[\sum_{i,t} w_{i,t}(\theta, \tau) \, \nabla_\theta \log \pi_\theta(o_{i,t} \mid q, o_{i,<t})\right], \tag{3}$$

where $w_{i,t}$ contains both the clipped importance ratio and the entropy-modulated advantage. A key distinction from PPO/GRPO is *global* normalization: MtPO normalizes token-level advantages across all sampled hypotheses such that $\sum_{i,t} w_{i,t} = 0$. This eliminates the verbosity bias introduced by per-sequence normalization, ensuring that long responses do not disproportionately influence gradient direction. The entropy-dependent scaling $(1 + \alpha H_t \gamma^t)$ further shapes the gradient landscape: high-entropy prefixes encourage broad exploration early in decoding, while the decay factor $\gamma^t$ gradually sharpens updates, improving stability without requiring a value critic.

**Constraint integration** RLVR (Eq. 1) introduces deterministic verifiers into MtPO's sampling process. For each prompt, $K$ candidates are generated from the old policy $\pi^\tau_{\theta_{\mathrm{old}}}$, scored by the composite reward $R_{\mathrm{RLVR}}$, and only the top-$G$ diverse and constraint-satisfying hypotheses are used for optimization. Because the verifiers operate solely on the output strings—via length ratios, markup preservation, language-ID checks, and code-mixing penalties—the filtering step is model-agnostic and computationally negligible. Importantly, determinism ensures that discarding invalid hypotheses does not introduce estimator bias: the gradient in Eq. 3 remains unbiased with respect to the feasible hypothesis space. This property aligns with the empirical 95% constraint-satisfaction rate reported in Table 2.

**Tokenizer efficiency** MtPO's entropy multiplier also provides a theoretical interpretation of vocabulary-adoption behavior under the expanded tokenizer. Let $N_{\mathrm{new}}$ denote the number of newly activated subword types during decoding. Then

$$\frac{\partial}{\partial t}\mathbb{E}[N_{\mathrm{new}}] = \alpha \, H_t \, \gamma^t \cdot \mathbb{E}[r^\tau_{i,t}(\theta)], \tag{4}$$

showing that early high-entropy states accelerate exploration over the expanded vocabulary, particularly for low-resource languages whose morphological units were newly introduced in Section 3. The decay factor $\gamma^t$ prevents uncontrolled sequence growth, ensuring that increased lexical diversity does not translate into verbosity. This aligns with the compression and latency improvements in Table 1 and the empirical trends in Figure 3.

Overall, MtPO forms a principled alignment framework whose gradient structure, verifier-based filtering, and entropy-driven vocabulary dynamics jointly yield stable, length-invariant, and constraint-aware multilingual optimization.

## 4 EXPERIMENTS

We validate MtPO across six dimensions: tokenization efficiency (Table 1, Section 4.1); token usage dynamics (Section 4.5); RLVR constraint verification (Section 4.3); ablations versus PPO, GRPO, DAPO, RLOO, and Reinforce++ under varied KL control (Section 4.4); length-control strategies with explicit penalties (Section 4.6); and overall translation and reasoning performance on 90 FLORES-Plus directions and four benchmarks (Section 4.7).

### 4.1 TOKENIZATION COMPARISON

**Tokenization efficiency.** To evaluate the effect of vocabulary expansion on low-resource languages, we compare the tokenization behavior of the original Qwen2.5-7B tokenizer with our Khmer-augmented tokenizer. Figure 2 presents a representative Khmer passage tokenized under both vocabularies. The baseline tokenizer fragments the text into 402 subwords due to insufficient coverage of Khmer morphemes, whereas our expanded tokenizer reduces this to 103 tokens by introducing language-specific units that capture common orthographic patterns.

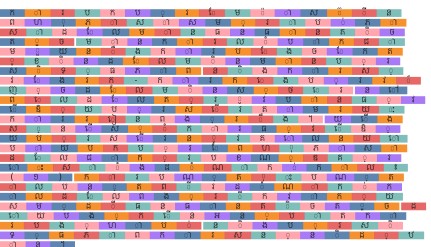 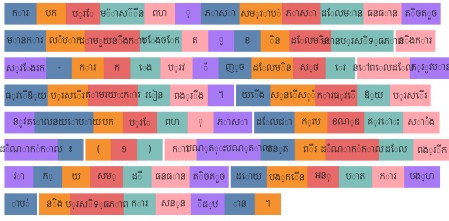

**(a)** Qwen2.5-7B tokenizer output (402 tokens)    **(b)** Khmer-augmented tokenizer output (103 tokens)

**Figure 2:** Tokenization comparison on a Khmer passage; Khmer-specific subwords curb over-segmentation and improve compression.

This case study illustrates the qualitative effect of targeted vocabulary expansion: low-resource languages benefit disproportionately from improved morphological coverage, resulting in shorter and more semantically coherent token sequences.

### 4.2 TOKEN EFFICIENCY ANALYSIS

**Corpus-level compression.** To quantify the effect of vocabulary expansion beyond single-passage case studies, we evaluate corpus-level tokenization efficiency on FLORES-Plus. Figure 3 reports output/input token ratios across nine translation directions, while Table 1 summarizes the vocabulary growth and compression gains for all expanded languages.

Across all settings, MtPO with the expanded tokenizer produces *consistently shorter* sequences than the baseline system. The effect is particularly pronounced for low-resource languages, where improved morphological coverage yields **3–5× compression gains** and thus directly reduces compute requirements during both training and inference.

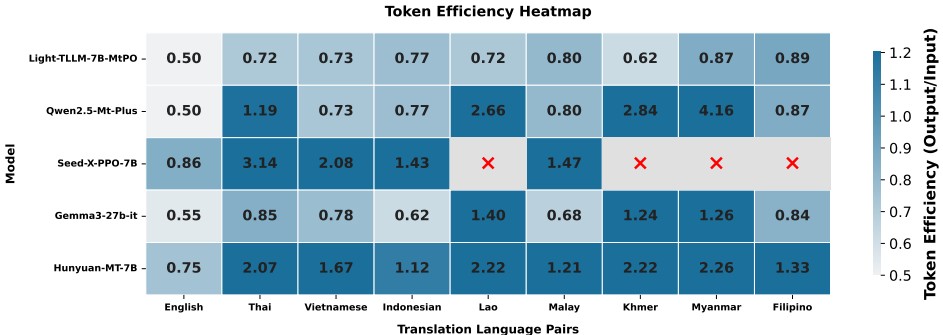

**Figure 3:** Output/input token ratios on FLORES-Plus; the expanded tokenizer yields the shortest sequences across nine language pairs.

Taking Figures 2 and 3 together with Table 1, MtPO's vocabulary expansion cuts Khmer tokens by about $4\times$, boosts FLORES-Plus compression ratios by up to $5.4\times$, and proportionally lowers training and inference compute—especially for the smallest corpora—while maintaining performance on high-resource languages.

### 4.3 CONSTRAINT VERIFICATION PERFORMANCE

We benchmark RLVR on four constraint checks (language, length, format, code mixing); the scores appear in Table 2.

Table 2 shows MtPO reaching 95.3% overall constraint accuracy, clearly ahead of general chat models and competitive translation systems, confirming RLVR's reliability.

**Table 1:** Tokenizer diagnostics for expanded low-resource languages (CR: tokens/char; ΔCR: absolute gain; **Speedup**: multiplicative gain; corpus size in billions).

| Language | Added tokens | Old CR | New CR | ΔCR | Speedup | Corpus size (B) |
|---|---|---|---|---|---|---|
| *Languages without additional tokens* | | | | | | |
| Filipino | – | 3.02 | 3.01 | -0.01 | 1.00 | 2.77 |
| Indonesian | – | 3.28 | 3.28 | 0.00 | 1.00 | 3.34 |
| Malay | – | 3.19 | 3.19 | 0.00 | 1.00 | 3.12 |
| Vietnamese | – | 3.40 | 3.40 | 0.00 | 1.00 | 3.67 |
| *Languages with MtPO vocabulary expansion* | | | | | | |
| Khmer | 3712 | 0.85 | 3.49 | 2.64 | **4.09** | 0.92 |
| Lao | 3359 | 0.85 | 3.05 | 2.20 | **3.59** | 0.26 |
| Mongolian | 4240 | 0.78 | 2.79 | 2.01 | **3.59** | 1.45 |
| Myanmar | 3226 | 0.69 | 2.87 | 2.18 | **4.17** | 1.58 |
| Tamil | 3942 | 0.93 | 2.85 | 1.92 | **3.07** | 2.97 |
| Thai | 2958 | 1.79 | 2.97 | 1.18 | **1.66** | 2.76 |
| Tibetan | 3920 | 0.75 | 4.03 | 3.28 | **5.39** | 0.17 |
| Uyghur | 3524 | 1.38 | 2.46 | 1.08 | **1.79** | 0.23 |

**Table 2:** RLVR constraint verification performance across different models. Best scores in bold.

| Model | Lang. | Length | Format | Mixing | Overall |
|---|---|---|---|---|---|
| Light-TLLM-7B-MtPO | **97.8** | 99.2 | **92.15** | 92.3 | **95.3** |
| Qwen2.5-7B-Instruct | 92.0 | 97.0 | 51.8 | 62.8 | 75.9 |
| Gemma3-27B-IT | 97.4 | 91.6 | 42.1 | 90.9 | 80.5 |
| Qwen-MT-Plus | 97.6 | **99.8** | 82.5 | 94.8 | 93.6 |
| Seed-X-PPO-7B | 97.6 | 79.8 | 79.0 | 90.3 | 86.6 |
| DeepSeek-V3 | 95.4 | 95.7 | 67.6 | 95.0 | 88.4 |
| Hunyuan-MT-7B | 91.8 | 90.7 | 71.1 | **96.2** | 87.4 |

## 4.4 POLICY OPTIMIZATION ALGORITHM COMPARISON

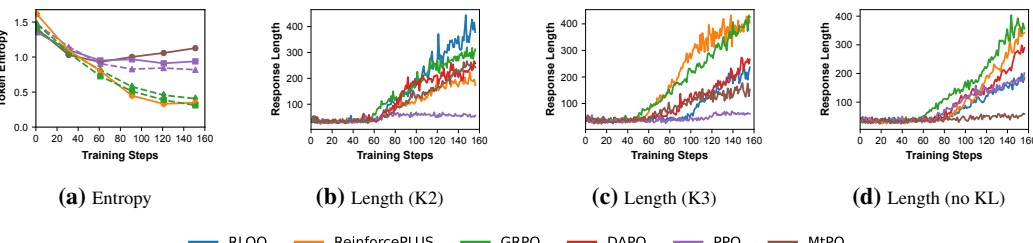

(a) Entropy     (b) Length (K2)     (c) Length (K3)     (d) Length (no KL)

RLOO   ReinforcePLUS   GRPO   DAPO   PPO   MtPO

**Figure 4:** Six RL algorithms across three KL regimes (18 runs).

For the ablations, we benchmark MtPO against PPO (Schulman et al., 2017), GRPO (Liu et al., 2025), DAPO (Liu et al., 2024), RLOO, and Reinforce++ (Hu et al., 2025) under K2, K3, and no-KL configurations (18 runs total) to disentangle the contributions of global length normalization, entropy-aware advantages, and asymmetric clipping, with entropy and response-length diagnostics as the primary endpoints.

Figure 4 presents eighteen runs spanning six policy-optimization algorithms (PPO, GRPO, DAPO, RLOO, Reinforce++, MtPO) evaluated under the K2, K3, and no-KL regimes from Section 3. K2 uses the moving-target KL controller from Reinforce++ and DAPO, K3 enforces a fixed KL budget, and the no-KL condition isolates stability when only entropy regularization is active. MtPO matches PPO-like stability while avoiding the failure modes of other critic-free methods. In line with de Oliveira et al. (2025), GRPO's lack of a critic and sequence-level normalization leads to two pathologies—rapid entropy collapse and uncontrolled response-length growth—which we also observe: baselines either collapse under K3 or inflate lengths under no-KL.

MtPO mitigates these instabilities via the mechanisms in Section 3: a position-aware entropy schedule that decays with depth and temperature-consistent importance ratios that align sampling and optimization distributions. The schedule preserves high exploration entropy early while tightening later, and the ratios prevent explosion when KL control is relaxed. Consequently, MtPO maintains high entropy and stable lengths across all KL settings—surpassing PPO in late-stage entropy retention. Because RLVR enforces length, markup, and language constraints deterministically, these results justify deploying the entropy-regularized no-KL configuration in practice: MtPO uses the full KL budget for lexical diversity, avoids critic-related instability, eliminates KL-target tuning, and remains robust even under the stricter K2/K3 regimes.

## 4.5 TOKEN USAGE TRAINING DYNAMICS

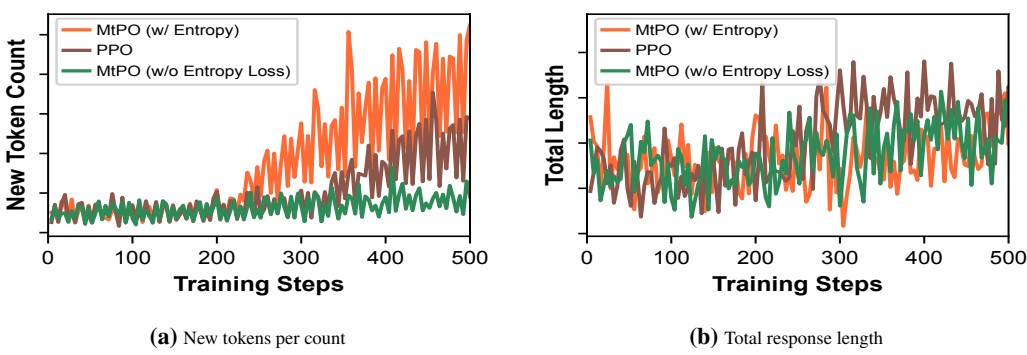

**(a)** New tokens per count

**(b)** Total response length

**Figure 5:** MtPO adopts new vocabulary faster than PPO while keeping total response length stable.

Figure 5 tracks new token usage and response length during MtPO training compared to standard PPO. MtPO uses more new tokens without length inflation. We inspect how the expanded vocabulary affects training dynamics, tracking new-token usage and response length during MtPO training versus PPO.

## 4.6 LENGTH CONTROL STRATEGY ANALYSIS

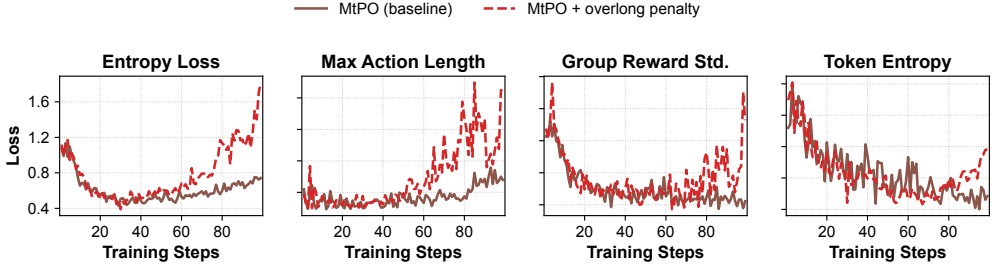

**Figure 6:** MtPO with/without overlength penalty. The penalty shortens outputs but destabilizes training.

Figure 6 isolates the effect of the DAPO-style overlength penalty by comparing MtPO runs with and without the additional term. Although the penalty shortens outputs, it provokes sharp entropy oscillations, intermittent reward spikes, and eventual collapse. We therefore depend on RLVR structural checks for length control, avoiding direct length penalties that compromise training stability.

## 4.7 OVERALL PERFORMANCE

We evaluate MtPO on FLORES-Plus (90 ASEAN language directions) using sacreBLEU, COMET, and chrF, and on general benchmarks (BBH, CMMLU, HellaSwag, MMLU), all with the Light-TLLM-7B family and the tokenizer/training recipe from Section 3.

MtPO achieves the strongest overall performance across all translation directions, with particularly impressive gains on en→xx translation (32.7 BLEU) where our method provides a 1.1 BLEU im-

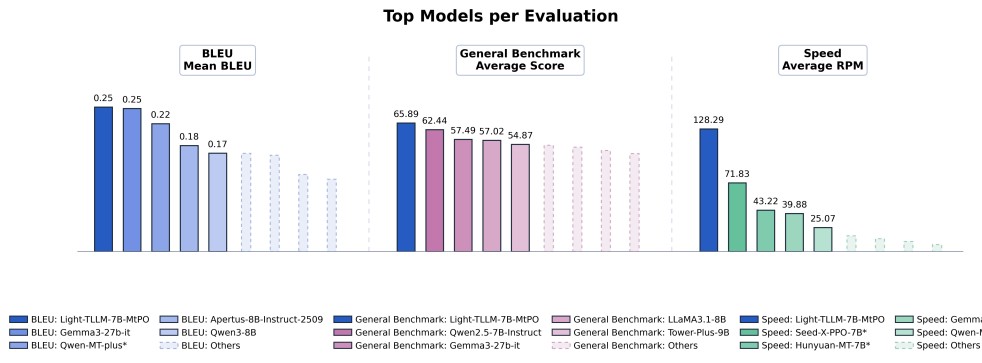

| | Translation (sacreBLEU/COMET/chrF) | | | | General Capabilities (%) | | | |
|---|---|---|---|---|---|---|---|---|
| Model | xx→en | en→xx | xx→xx | Avg. | BBH | CMMLU | HellaSwag | MMLU |
| *Multilingual Chat Models* | | | | | | | | |
| Gemma3-27B-IT | 36.8/0.882/62.87 | 30.7/0.875/54.15 | 22.3/0.847/47.53 | 24.7/0.854/49.73 | 55.9 | 55.9 | 55.9 | **56.0** |
| Qwen3-8B | 31.1/0.862/58.45 | 23.3/0.799/46.42 | 14.4/0.752/36.80 | 16.9/0.767/39.93 | **63.8** | 60.8 | 26.0 | 51.3 |
| Qwen2.5-7B-Instruct | 24.8/0.498/19.41 | 17.4/0.455/15.58 | 9.2/0.449/14.50 | 11.6/0.454/15.10 | 54.4 | **64.1** | **85.2** | 40.9 |
| Apertus-8B-Instruct | 32.5/0.870/60.51 | 25.7/0.802/46.63 | 15.6/0.750/36.86 | 18.3/0.767/40.22 | 49.2 | 45.3 | 64.2 | 45.2 |
| Tower-Plus-9B | 28.2/0.825/54.73 | 18.3/0.671/37.74 | 9.8/0.615/29.33 | 12.5/0.641/32.74 | 40.4 | 57.2 | 73.1 | 42.1 |
| *Translation-Focused Models* | | | | | | | | |
| Qwen-MT-Plus | 34.0/0.881/60.10 | 29.6/0.869/53.44 | 19.6/0.839/45.79 | 22.1/0.846/48.19 | – | – | – | – |
| Seed-X-PPO-7B | 25.9/0.786/51.61 | 22.6/0.708/36.33 | 10.5/0.638/24.72 | 13.3/0.660/28.57 | – | – | – | – |
| Hunyuan-MT-7B | 24.6/0.839/55.24 | 23.4/0.862/48.38 | 14.8/0.802/39.53 | 16.6/0.812/41.99 | – | – | – | – |
| *Translation-Focused without LLM Models* | | | | | | | | |
| Google Translate | **41.2/0.884/65.68** | 32.2/0.842/55.70 | **23.2**/0.820/47.94 | **25.9**/0.828/50.49 | – | – | – | – |
| NLLB-200 | 38.1/0.875/62.95 | 28.3/0.845/52.13 | 19.7/0.830/44.92 | 22.4/0.836/47.45 | – | – | – | – |
| *Our Models* | | | | | | | | |
| Light-TLLM-7B-MtPO w/o CPT | 33.3/0.862/56.82 | 31.7/0.863/51.82 | 21.4/0.822/43.60 | 23.7/0.837/46.86 | 59.7 | 62.0 | 83.5 | 47.6 |
| Δ (MtPO − w/o CPT) | +2.8/0.019/+5.92 | +1.0/0.019/+4.40 | +1.7/+0.032/+5.20 | +1.2/+0.022/+4.08 | +1.2 | +1.2 | +1.7 | +0.9 |
| Light-TLLM-7B-SFT | 35.4/0.875/59.82 | 32.0/0.875/52.94 | 22.7/0.839/44.46 | 24.3/0.849/48.26 | 59.6 | 61.4 | 83.7 | 47.2 |
| Δ (MtPO − SFT) | +0.7/+0.006/+2.92 | +0.7/+0.007/+3.28 | +0.4/+0.015/+4.34 | +0.6/+0.010/+2.68 | +1.3 | +1.8 | +1.5 | +1.3 |
| Light-TLLM-7B-MtPO | 36.1/0.881/62.74 | **32.7/0.882/56.22** | 23.1/**0.854/48.80** | 24.9/**0.859/50.94** | 60.9 | 63.2 | **85.2** | 48.5 |

**Table 3:** Overall performance comparison on translation and instruction-following benchmarks.

provement over the next best system. The results demonstrate several key advantages: **(1) Multilingual gains:** Significant improvements on xx→xx translation (23.1 BLEU), which is particularly challenging due to limited parallel training data. **(2) Capability preservation:** Strong performance on general reasoning benchmarks (BBH: 60.9%, CMMLU: 63.2%, HellaSwag: 85.2%, MMLU: 48.5%) shows that translation specialization does not compromise broad capabilities. **(3) Efficiency:** Our model achieves competitive performance with 7B parameters compared to much larger models like Gemma3-27B-IT. The consistent improvements across diverse language pairs and task types validate the effectiveness of our unified approach to multilingual translation optimization. Detailed per-language BLEU/COMET/chrF scores are provided in Appendices A.6, A.7, and A.8.

Training configurations and evaluation metrics are detailed in Appendix A.5.

## 5 CONCLUSION

In this work, we have presented a comprehensive approach to extending foundation models to low-resource languages through Multilingual Translation Policy Optimization (MtPO). Our methodology addresses three critical challenges in multilingual model development: tokenization efficiency, balanced multilingual training, and effective reinforcement learning for translation tasks. In addition, our Reinforcement Learning with Verifiable Rewards (RLVR) adds deterministic checks on length ratio, structural tokens, language targeting, and mixing to substantially reduce real-world failure modes without harming general translation quality.

### ACKNOWLEDGMENTS

We also acknowledge the use of large language models (LLMs) for text polishing and writing assistance during the preparation of this manuscript.

## ETHICS STATEMENT

This research focuses on improving machine translation for low-resource languages, particularly Southeast Asian languages. Our work aims to reduce language barriers and promote linguistic diversity, which has positive societal impacts. The research does not involve human subjects, sensitive data collection, or potential harmful applications. All datasets used are publicly available, and our methods are designed to benefit underrepresented language communities. We acknowledge that improved translation capabilities could potentially be misused, but we believe the benefits of enabling better cross-cultural communication and preserving linguistic diversity significantly outweigh potential risks.

## REPRODUCIBILITY STATEMENT

We are committed to ensuring the reproducibility of our results. All experimental details, hyperparameters, and training configurations are provided in Appendix A.5. The datasets used (FLORES-Plus, BBH, CMMLU, HellaSwag, MMLU) are publicly available. Model architectures, training procedures, and evaluation protocols are described in detail throughout the paper. Upon acceptance, we plan to release our code, model checkpoints, and detailed training scripts to facilitate reproduction of our results. The tokenizer expansion methodology and RLVR constraint verification framework are documented with sufficient detail to enable replication.

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

## A  SUPPLEMENTARY MATERIAL

### A.1  TOKENIZER EXPANSION DETAILS

Table 1 in Section 4.1 reports detailed tokenizer diagnostics for the expanded low-resource languages, showing compression ratios and efficiency improvements achieved through vocabulary expansion.

### A.2  DETAILED RLVR FORMULATIONS

The Reward Learning with Verifiable Rewards (RLVR) framework combines semantic translation quality signals with deterministic, verifiable constraints. The complete reward function is defined as:

$$R_{\text{RLVR}}(x, y) = R_{\text{mt}}(x, y) + \lambda_{\text{len}} \, r_{\text{len}}(x, y) + \lambda_{\text{fmt}} \, r_{\text{fmt}}(x, y)$$
$$+ \lambda_{\text{lid}} \, r_{\text{lid}}(y) + \lambda_{\text{mix}} \, r_{\text{mix}}(y) \tag{5}$$

where $R_{\text{mt}}$ represents the semantic translation reward and each verifiable term is defined as follows:

### A.2.1 LENGTH RATIO REWARD

The length ratio reward $r_{\text{len}}(x, y)$ constrains output length relative to input:

$$r_{\text{len}}(x, y) = \begin{cases} +1, & \text{if } \rho \in [\alpha, \beta] \\ -\sigma_{\text{len}} \cdot \max(0, \rho - \beta), & \text{if } \rho > \beta \\ -\sigma_{\text{len}} \cdot \max(0, \alpha - \rho), & \text{if } \rho < \alpha \end{cases} \tag{6}$$

where $\rho = \frac{|y|}{|x|}$ is the length ratio, $[\alpha, \beta]$ defines the acceptable ratio range (typically $[0.5, 2.0]$ for most language pairs), and $\sigma_{\text{len}} > 0$ controls the penalty strength for violations.

### A.2.2 FORMAT VALIDITY REWARD

For inputs containing structural tokens (HTML, Markdown, etc.), the format validity reward $r_{\text{fmt}}(x, y)$ is computed as:

$$r_{\text{fmt}}(x, y) = w_{\text{preserve}} \cdot f_{\text{preserve}}(x, y) - w_{\text{broken}} \cdot f_{\text{broken}}(y) \tag{7}$$

where:

- $f_{\text{preserve}}(x, y) = \frac{|\text{struct\_tokens}(x) \cap \text{struct\_tokens}(y)|}{|\text{struct\_tokens}(x)|}$ measures the preservation ratio of structural tokens

- $f_{\text{broken}}(y)$ counts syntax violations (unclosed tags, broken lists, malformed code fences) detected by lightweight parsers

- $w_{\text{preserve}}, w_{\text{broken}} > 0$ are weighting factors

### A.2.3 TARGET LANGUAGE VERIFICATION

The language identification reward $r_{\text{lid}}(y)$ verifies that the output matches the target language $\ell_t$:

$$r_{\text{lid}}(y) = \begin{cases} +1, & \text{if } \text{LID}(y) = \ell_t \text{ and confidence} > \theta_{\text{lid}} \\ -\eta_{\text{lid}}, & \text{otherwise} \end{cases} \tag{8}$$

where $\text{LID}(\cdot)$ is a language identification model, $\theta_{\text{lid}}$ is the confidence threshold (typically 0.8), and $\eta_{\text{lid}} > 0$ is the penalty for off-target language.

### A.2.4 CODE-MIXING DETECTION

The code-mixing reward $r_{\text{mix}}(y)$ penalizes excessive mixing of languages within the output:

$$r_{\text{mix}}(y) = \begin{cases} 0, & \text{if } p_{\text{mix}}(y) \leq \tau_{\text{mix}} \\ -\zeta_{\text{mix}} \cdot (p_{\text{mix}}(y) - \tau_{\text{mix}}), & \text{otherwise} \end{cases} \tag{9}$$

where $p_{\text{mix}}(y)$ estimates the proportion of non-$\ell_t$ segments or scripts in $y$, $\tau_{\text{mix}}$ is the mixing tolerance threshold (typically 0.1-0.2), and $\zeta_{\text{mix}} > 0$ controls the penalty severity.

### A.2.5 IMPLEMENTATION DETAILS

All verifiable terms are designed to be computationally efficient and deterministic:

- **Clipping**: Each reward term is clipped to $[-C_{\max}, C_{\max}]$ to prevent gradient explosions, where $C_{\max} = 5.0$ in our implementation.
- **Normalization**: The final reward is normalized across the batch to maintain stable training dynamics.
- **Hyperparameters**: We use $\lambda_{\text{len}} = 0.3$, $\lambda_{\text{fmt}} = 0.2$, $\lambda_{\text{lid}} = 0.4$, $\lambda_{\text{mix}} = 0.3$ as default weights, tuned on validation sets.

The RLVR framework thus provides structured, interpretable constraints that complement learned semantic rewards while maintaining computational efficiency during training.

### A.3 REWARD MODEL ERROR CATEGORIES

In our reward model design, we systematically categorized translation errors into ten distinct types to ensure comprehensive coverage of potential failure modes. This categorization framework was essential for training a robust reward model that could effectively identify and appropriately score various types of translation errors across multiple languages.

**Table 4:** Error categories for RM robustness

| Category[1] | Type | Description |
| --- | --- | --- |
| Quality | Accuracy | Mistranslation, omission, over-translation, semantic misunderstanding |
| | Off-target | Untranslated segments, wrong target language |
| | Fluency | Natural expression, comprehensibility, professionalism |
| | Terminology | Consistency, accuracy, omission of terms |
| | Code-mixing | Unreasonable language mixing within sentences |
| | Repetition | End-of-sentence, segment, instruction repetition |
| Instruction | Intent | Machine misunderstanding of instructions |
| | Leakage | Output containing or translating instructions |
| Content | Explanation | Excessive explanations beyond translation |
| Format[2] | Preservation | Maintaining original formatting and punctuation |

[1] Categories are non-exclusive; a sample may trigger multiple error types.
[2] Preservation includes punctuation, special markers, and structural tokens.

This comprehensive error categorization system enabled our reward model to provide nuanced feedback on translation quality, distinguishing between different types of errors and their relative severity. The framework was particularly important for low-resource languages where translation errors can manifest in diverse and subtle ways.

### A.4 DETAILED MtPO MATHEMATICAL DERIVATIONS

This section provides comprehensive mathematical details for the MtPO formulation presented in the main text.

### A.4.1 ADVANTAGE ESTIMATION AND NORMALIZATION

The MtPO advantage estimator is constructed as:

$$\hat{A}_{i,t} = \frac{R_{i,t} - B_t(q)}{\text{std}_{\text{microbatch}}(\{R_{j,t}\}_{j=1}^G) + \epsilon} \cdot (1 + \alpha \cdot H(\pi_\theta(o_{i,t}|q, o_{i,<t})) \cdot \gamma^t) \tag{10}$$

where $B_t(q) = \text{mean}_{\text{group}}(\{R_{j,t}\}_{j=1}^G)$ is the groupwise baseline. This construction ensures:

- **Zero-mean property**: $\sum_{i=1}^G \hat{A}_{i,t} = 0$ almost surely for every position $t$, preserving the fixed-point of policy gradient updates.

- **Scale invariance**: Microbatch standardization aligns estimator scale across batches, crucial for stable clipping in multilingual settings where reward magnitudes vary substantially.

- **Position-aware exploration**: The entropy multiplier $1 + \alpha H(\pi_\theta(o_{i,t}|q, o_{i,<t}))\gamma^t$ amplifies early token contributions, reflecting their larger causal influence on sequence completion.

### A.4.2 DUAL ENTROPY REGULARIZATION

MtPO employs two complementary entropy terms:

$$\text{Local entropy : within } \hat{A}_{i,t} \text{ for credit assignment modulation} \tag{11}$$

$$\text{Global entropy : } \beta \mathbb{E}[H(\pi_\theta(\cdot|q, o_{i,<t}))] \text{ for exploration enforcement} \tag{12}$$

The local entropy term modulates credit assignment without introducing high-variance gradients, while the global term enforces a lower bound on policy entropy. This dual structure prevents premature action distribution collapse observed in non-value baselines.

### A.4.3 TEMPERATURE-CONSISTENT IMPORTANCE SAMPLING

The importance ratio admits the closed-form expression:

$$r_{i,t}^\tau(\theta) = \exp\left(\frac{\log \pi_\theta(o_{i,t}|q, o_{i,<t}) - \log \pi_{\theta_{\text{old}}}(o_{i,t}|q, o_{i,<t})}{\tau}\right) \tag{13}$$

This formulation preserves the martingale property required for on-policy convergence analyses while ensuring consistent temperature scaling across data collection and policy updates.

### A.4.4 ASYMMETRIC CLIPPING ANALYSIS

The asymmetric clipping bounds $(1 - \epsilon_{\text{low}}, 1 + \epsilon_{\text{high}})$ with $\epsilon_{\text{low}} > \epsilon_{\text{high}}$ reflect empirical observations:

- **Downward sensitivity**: Ratio decreases (policy becoming less likely to generate observed tokens) correlate strongly with translation quality degradation.

- **Upward tolerance**: Moderate ratio increases often correspond to improved translation fluency and adequacy.

- **Typical values**: $\epsilon_{\text{low}} = 0.20$, $\epsilon_{\text{high}} = 0.28$ based on extensive hyperparameter search.

### A.4.5 COMPUTATIONAL OPTIMIZATIONS

For large vocabularies, entropy computation is approximated by:

$$H(\pi_\theta(\cdot|s)) \approx -\sum_{k \in \text{top-20\%}} \pi_\theta(k|s) \log \pi_\theta(k|s) \tag{14}$$

This approximation reduces computational overhead by 80% while maintaining sufficient accuracy for gradient-based training, as the tail of the probability distribution contributes negligibly to entropy estimates.

## A.5 EXPERIMENTAL SETUP DETAILS

**Models.** We compare three checkpoints derived from our pipeline—continued pre-training only (SFT W/O CPT), continued pre-training followed by instruction tuning (SFT), and the full MtPO stage—against both multilingual chat models (Gemma3, Qwen2.5/3, Aya, LLaMAX, etc.) and translation-specialized systems (Hunyuan-MT-7B, Qwen-MT-Plus, Seed-X-PPO-7B).

**Training budget.** Continued pre-training consumes 200B tokens with adaptive multilingual sampling, while MtPO uses 60k RL steps with per-step batch size of 128 samples. All RL baselines share identical budgets to enable fair comparisons.

**Metrics.** Main results report sacreBLEU for three translation groups (xx → en, en → xx, and xx → xx). General capabilities are measured with zero-shot accuracy. Significance is assessed with paired bootstrap resampling across FLORES directions.

## A.6 DETAILED BLEU BREAKDOWN

Table 5 reports the BLEU scores by source and target language pair across all evaluated models, computed with the standard n-gram formulation(Papineni et al., 2002). Language names from the original dataset are translated into English for readability.

**Table 5:** BLEU scores per source–target language pair.

| Source | Target | Model BLEU Scores | | | | | | | | | | | | | | | |
|--------|--------|------|-------|--------|------|------|-------|------|-------|------|------|--------|-------|-------|-------|--------|------|
| | | Gem3 | SeedX | QwenMT | Huny | MtPO | Qw2.5 | Qw3 | Apert | Aya | Emma | LLa3.1 | LLaX3 | Mistr | Tower | Google | NLLB |
| Thai | Malay | 0.247 | 0.250 | 0.212 | 0.180 | 0.243 | 0.132 | 0.169 | 0.187 | 0.092 | 0.008 | 0.014 | 0.173 | 0.031 | 0.131 | 0.234 | 0.227 |
| Thai | Khmer | 0.078 | 0.006 | 0.051 | 0.090 | 0.091 | 0.018 | 0.042 | 0.005 | 0.002 | 0.077 | 0.012 | 0.063 | 0.002 | 0.003 | 0.095 | 0.072 |
| Thai | Lao | 0.174 | 0.012 | 0.133 | 0.007 | 0.199 | 0.010 | 0.067 | 0.026 | 0.003 | 0.006 | 0.005 | 0.066 | 0.003 | 0.005 | 0.147 | 0.174 |
| Thai | Burmese | 0.102 | 0.002 | 0.135 | 0.121 | 0.161 | 0.013 | 0.048 | 0.095 | 0.001 | 0.067 | 0.015 | 0.047 | 0.003 | 0.002 | 0.116 | 0.066 |
| Thai | Filipino | 0.234 | 0.020 | 0.179 | 0.156 | 0.214 | 0.066 | 0.130 | 0.169 | 0.054 | 0.029 | 0.091 | 0.149 | 0.026 | 0.092 | 0.148 | 0.211 |
| Thai | Indonesian | 0.302 | 0.277 | 0.197 | 0.195 | 0.259 | 0.191 | 0.232 | 0.250 | 0.160 | 0.008 | 0.033 | 0.204 | 0.047 | 0.192 | 0.285 | 0.266 |
| Thai | Vietnamese | 0.318 | 0.321 | 0.240 | 0.234 | 0.296 | 0.212 | 0.268 | 0.278 | 0.190 | 0.015 | 0.030 | 0.256 | 0.056 | 0.210 | 0.321 | 0.294 |
| Thai | English | 0.327 | 0.298 | 0.262 | 0.231 | 0.312 | 0.274 | 0.291 | 0.282 | 0.168 | 0.061 | 0.037 | 0.273 | 0.145 | 0.287 | 0.325 | 0.316 |
| Thai | Chinese | 0.266 | 0.184 | 0.260 | 0.214 | 0.243 | 0.217 | 0.241 | 0.231 | 0.137 | 0.053 | 0.111 | 0.176 | 0.072 | 0.234 | 0.281 | 0.147 |
| Malay | Thai | 0.292 | 0.291 | 0.287 | 0.246 | 0.269 | 0.162 | 0.227 | 0.243 | 0.060 | 0.004 | 0.192 | 0.206 | 0.030 | 0.145 | 0.321 | 0.202 |
| Malay | Khmer | 0.086 | 0.006 | 0.053 | 0.103 | 0.103 | 0.019 | 0.044 | 0.006 | 0.003 | 0.017 | 0.013 | 0.070 | 0.004 | 0.004 | 0.105 | 0.083 |
| Malay | Lao | 0.183 | 0.015 | 0.144 | 0.009 | 0.231 | 0.016 | 0.061 | 0.028 | 0.006 | 0.005 | 0.006 | 0.073 | 0.005 | 0.008 | 0.168 | 0.197 |
| Malay | Burmese | 0.118 | 0.003 | 0.146 | 0.134 | 0.174 | 0.016 | 0.056 | 0.107 | 0.001 | 0.021 | 0.016 | 0.050 | 0.004 | 0.002 | 0.115 | 0.079 |
| Malay | Filipino | 0.259 | 0.027 | 0.229 | 0.208 | 0.273 | 0.079 | 0.159 | 0.203 | 0.099 | 0.028 | 0.177 | 0.207 | 0.050 | 0.125 | 0.048 | 0.274 |
| Malay | Indonesian | 0.385 | 0.374 | 0.366 | 0.270 | 0.312 | 0.239 | 0.307 | 0.263 | 0.336 | 0.080 | 0.254 | 0.254 | 0.113 | 0.239 | 0.381 | 0.368 |
| Malay | Vietnamese | 0.365 | 0.365 | 0.350 | 0.285 | 0.337 | 0.252 | 0.304 | 0.323 | 0.315 | 0.011 | 0.288 | 0.294 | 0.086 | 0.250 | 0.384 | 0.340 |
| Malay | English | 0.440 | 0.438 | 0.441 | 0.341 | 0.435 | 0.373 | 0.397 | 0.413 | 0.379 | 0.079 | 0.363 | 0.402 | 0.274 | 0.421 | 0.504 | 0.466 |
| Malay | Chinese | 0.280 | 0.185 | 0.280 | 0.259 | 0.262 | 0.227 | 0.255 | 0.244 | 0.224 | 0.059 | 0.203 | 0.189 | 0.101 | 0.250 | 0.329 | 0.167 |
| Khmer | Thai | 0.265 | 0.043 | 0.228 | 0.183 | 0.243 | 0.090 | 0.089 | 0.179 | 0.015 | 0.013 | 0.090 | 0.180 | 0.014 | 0.072 | 0.272 | 0.177 |
| Khmer | Malay | 0.237 | 0.054 | 0.200 | 0.155 | 0.234 | 0.070 | 0.141 | 0.171 | 0.032 | 0.093 | 0.005 | 0.160 | 0.017 | 0.072 | 0.245 | 0.239 |
| Khmer | Lao | 0.141 | 0.010 | 0.100 | 0.008 | 0.209 | 0.009 | 0.009 | 0.001 | 0.006 | 0.013 | 0.005 | 0.059 | 0.006 | 0.006 | 0.149 | 0.176 |
| Khmer | Burmese | 0.090 | 0.002 | 0.116 | 0.109 | 0.153 | 0.002 | 0.030 | 0.002 | 0.002 | 0.073 | 0.013 | 0.043 | 0.002 | 0.001 | 0.104 | 0.092 |
| Khmer | Filipino | 0.229 | 0.015 | 0.174 | 0.141 | 0.221 | 0.048 | 0.104 | 0.159 | 0.033 | 0.011 | 0.018 | 0.148 | 0.015 | 0.054 | 0.217 | 0.216 |
| Khmer | Indonesian | 0.279 | 0.062 | 0.238 | 0.168 | 0.260 | 0.096 | 0.183 | 0.223 | 0.044 | 0.082 | 0.008 | 0.180 | 0.022 | 0.095 | 0.291 | 0.266 |
| Khmer | Vietnamese | 0.296 | 0.069 | 0.265 | 0.195 | 0.295 | 0.112 | 0.214 | 0.249 | 0.038 | 0.049 | 0.004 | 0.233 | 0.022 | 0.106 | 0.313 | 0.288 |
| Khmer | English | 0.315 | 0.066 | 0.294 | 0.202 | 0.322 | 0.128 | 0.236 | 0.273 | 0.044 | 0.015 | 0.129 | 0.266 | 0.048 | 0.134 | 0.335 | 0.332 |
| Khmer | Chinese | 0.238 | 0.014 | 0.211 | 0.165 | 0.227 | 0.095 | 0.175 | 0.180 | 0.050 | 0.053 | 0.027 | 0.153 | 0.022 | 0.113 | 0.262 | 0.143 |
| Lao | Thai | 0.297 | 0.016 | 0.281 | 0.038 | 0.275 | 0.085 | 0.134 | 0.093 | 0.016 | 0.041 | 0.110 | 0.201 | 0.007 | 0.092 | 0.316 | 0.215 |
| Lao | Malay | 0.275 | 0.016 | 0.222 | 0.039 | 0.284 | 0.062 | 0.164 | 0.191 | 0.037 | 0.109 | 0.015 | 0.144 | 0.010 | 0.076 | 0.280 | 0.267 |
| Lao | Khmer | 0.087 | 0.002 | 0.042 | 0.041 | 0.109 | 0.009 | 0.011 | 0.002 | 0.004 | 0.053 | 0.009 | 0.065 | 0.005 | 0.004 | 0.103 | 0.084 |
| Lao | Burmese | 0.103 | 0.001 | 0.105 | 0.039 | 0.165 | 0.003 | 0.026 | 0.002 | 0.002 | 0.039 | 0.009 | 0.035 | 0.003 | 0.002 | 0.113 | 0.095 |
| Lao | Filipino | 0.246 | 0.005 | 0.188 | 0.051 | 0.239 | 0.049 | 0.127 | 0.167 | 0.035 | 0.050 | 0.054 | 0.129 | 0.012 | 0.048 | 0.244 | 0.235 |
| Lao | Indonesian | 0.314 | 0.018 | 0.228 | 0.040 | 0.292 | 0.072 | 0.196 | 0.231 | 0.047 | 0.067 | 0.042 | 0.157 | 0.014 | 0.087 | 0.333 | 0.296 |
| Lao | Vietnamese | 0.329 | 0.023 | 0.270 | 0.039 | 0.314 | 0.087 | 0.229 | 0.259 | 0.040 | 0.057 | 0.032 | 0.217 | 0.013 | 0.096 | 0.350 | 0.301 |
| Lao | English | 0.359 | 0.014 | 0.257 | 0.048 | 0.364 | 0.094 | 0.266 | 0.282 | 0.051 | 0.056 | 0.120 | 0.236 | 0.029 | 0.120 | 0.415 | 0.375 |
| Lao | Chinese | 0.247 | 0.005 | 0.235 | 0.050 | 0.231 | 0.072 | 0.184 | 0.189 | 0.054 | 0.055 | 0.030 | 0.126 | 0.013 | 0.102 | 0.295 | 0.150 |
| Burmese | Thai | 0.228 | 0.015 | 0.203 | 0.145 | 0.204 | 0.046 | 0.040 | 0.157 | 0.005 | 0.013 | 0.080 | 0.123 | 0.007 | 0.036 | 0.249 | 0.162 |
| Burmese | Malay | 0.194 | 0.017 | 0.170 | 0.128 | 0.193 | 0.031 | 0.099 | 0.141 | 0.009 | 0.093 | 0.004 | 0.092 | 0.005 | 0.051 | 0.212 | 0.220 |
| Burmese | Khmer | 0.055 | 0.002 | 0.055 | 0.068 | 0.073 | 0.007 | 0.013 | 0.003 | 0.002 | 0.041 | 0.005 | 0.040 | 0.002 | 0.001 | 0.079 | 0.062 |
| Burmese | Lao | 0.113 | 0.004 | 0.092 | 0.004 | 0.169 | 0.003 | 0.015 | 0.010 | 0.003 | 0.023 | 0.002 | 0.039 | 0.001 | 0.001 | 0.129 | 0.149 |
| Burmese | Filipino | 0.200 | 0.006 | 0.155 | 0.124 | 0.188 | 0.031 | 0.077 | 0.140 | 0.013 | 0.084 | 0.055 | 0.096 | 0.009 | 0.039 | 0.197 | 0.205 |
| Burmese | Indonesian | 0.233 | 0.018 | 0.210 | 0.132 | 0.201 | 0.046 | 0.132 | 0.181 | 0.013 | 0.086 | 0.009 | 0.109 | 0.008 | 0.070 | 0.259 | 0.249 |
| Burmese | Vietnamese | 0.260 | 0.020 | 0.235 | 0.166 | 0.254 | 0.057 | 0.164 | 0.218 | 0.014 | 0.057 | 0.020 | 0.173 | 0.010 | 0.077 | 0.286 | 0.269 |
| Burmese | English | 0.263 | 0.020 | 0.254 | 0.173 | 0.268 | 0.061 | 0.181 | 0.229 | 0.016 | 0.046 | 0.153 | 0.198 | 0.023 | 0.104 | 0.317 | 0.304 |
| Burmese | Chinese | 0.207 | 0.004 | 0.191 | 0.140 | 0.189 | 0.049 | 0.132 | 0.163 | 0.013 | 0.326 | 0.063 | 0.124 | 0.010 | 0.091 | 0.244 | 0.131 |
| Filipino | Thai | 0.298 | 0.246 | 0.280 | 0.226 | 0.267 | 0.134 | 0.205 | 0.218 | 0.051 | 0.008 | 0.173 | 0.197 | 0.025 | 0.125 | 0.320 | 0.238 |
| Filipino | Malay | 0.317 | 0.277 | 0.211 | 0.225 | 0.326 | 0.131 | 0.203 | 0.193 | 0.146 | 0.095 | 0.180 | 0.229 | 0.052 | 0.171 | 0.376 | 0.422 |
| Filipino | Khmer | 0.083 | 0.006 | 0.060 | 0.095 | 0.103 | 0.016 | 0.038 | 0.005 | 0.003 | 0.011 | 0.010 | 0.071 | 0.004 | 0.003 | 0.123 | 0.090 |
| Filipino | Lao | 0.164 | 0.015 | 0.136 | 0.008 | 0.227 | 0.013 | 0.046 | 0.023 | 0.005 | 0.558 | 0.006 | 0.064 | 0.004 | 0.005 | 0.197 | 0.232 |
| Filipino | Burmese | 0.122 | 0.003 | 0.128 | 0.132 | 0.168 | 0.015 | 0.050 | 0.093 | 0.002 | 0.019 | 0.017 | 0.050 | 0.004 | 0.002 | 0.128 | 0.122 |
| Filipino | Indonesian | 0.375 | 0.300 | 0.263 | 0.231 | 0.336 | 0.176 | 0.229 | 0.246 | 0.245 | 0.161 | 0.221 | 0.261 | 0.078 | 0.203 | 0.332 | 0.362 |
| Filipino | Vietnamese | 0.368 | 0.315 | 0.139 | 0.264 | 0.347 | 0.194 | 0.279 | 0.294 | 0.254 | 0.056 | 0.268 | 0.285 | 0.067 | 0.204 | 0.499 | 0.467 |
| Filipino | English | 0.481 | 0.394 | 0.454 | 0.328 | 0.454 | 0.299 | 0.382 | 0.395 | 0.307 | 0.535 | 0.399 | 0.403 | 0.239 | 0.382 | 0.457 | 0.416 |
| Filipino | Chinese | 0.287 | 0.129 | 0.271 | 0.237 | 0.267 | 0.191 | 0.237 | 0.236 | 0.175 | 0.029 | 0.185 | 0.184 | 0.084 | 0.235 | 0.396 | 0.205 |
| Indonesian | Thai | 0.304 | 0.304 | 0.299 | 0.256 | 0.276 | 0.172 | 0.234 | 0.254 | 0.064 | 0.254 | 0.200 | 0.209 | 0.030 | 0.161 | 0.283 | 0.171 |
| Indonesian | Malay | 0.332 | 0.351 | 0.313 | 0.268 | 0.299 | 0.220 | 0.262 | 0.240 | 0.163 | 0.037 | 0.222 | 0.215 | 0.083 | 0.228 | 0.245 | 0.218 |
| Indonesian | Khmer | 0.085 | 0.006 | 0.064 | 0.102 | 0.101 | 0.018 | 0.045 | 0.006 | 0.003 | 0.119 | 0.012 | 0.078 | 0.004 | 0.004 | 0.085 | 0.065 |
| Indonesian | Lao | 0.178 | 0.015 | 0.138 | 0.008 | 0.226 | 0.016 | 0.060 | 0.030 | 0.006 | 0.047 | 0.006 | 0.067 | 0.005 | 0.007 | 0.135 | 0.138 |
| Indonesian | Burmese | 0.115 | 0.003 | 0.137 | 0.133 | 0.176 | 0.016 | 0.054 | 0.102 | 0.002 | 0.033 | 0.018 | 0.051 | 0.003 | 0.002 | 0.108 | 0.114 |

*Continued on next page*

| Source | Target | Model BLEU Scores | | | | | | | | | | | | | | | |
|---|---|---|---|---|---|---|---|---|---|---|---|---|---|---|---|---|---|---|
| | | Gem3 | SeedX | QwenMT | Huny | MtPO | Qw2.5 | Qw3 | Apert | Aya | Emma | LLa3.1 | LLaX3 | Mistr | Tower | Google | NLLB |
| Indonesian | Filipino | 0.262 | 0.025 | 0.236 | 0.209 | 0.273 | 0.085 | 0.170 | 0.211 | 0.101 | 0.081 | 0.181 | 0.211 | 0.049 | 0.124 | 0.214 | 0.189 |
| Indonesian | Vietnamese | 0.371 | 0.377 | 0.362 | 0.296 | 0.349 | 0.270 | 0.321 | 0.341 | 0.350 | 0.491 | 0.310 | 0.307 | 0.087 | 0.270 | 0.299 | 0.241 |
| Indonesian | English | 0.447 | 0.443 | 0.446 | 0.344 | 0.435 | 0.399 | 0.411 | 0.428 | 0.416 | 0.012 | 0.350 | 0.395 | 0.269 | 0.435 | 0.334 | 0.292 |
| Indonesian | Chinese | 0.293 | 0.213 | 0.302 | 0.266 | 0.271 | 0.255 | 0.271 | 0.260 | 0.251 | 0.398 | 0.216 | 0.203 | 0.113 | 0.276 | 0.347 | 0.304 |
| Vietnamese | Thai | 0.280 | 0.252 | 0.277 | 0.232 | 0.254 | 0.155 | 0.222 | 0.237 | 0.065 | 0.285 | 0.184 | 0.200 | 0.029 | 0.149 | 0.300 | 0.198 |
| Vietnamese | Malay | 0.277 | 0.283 | 0.251 | 0.215 | 0.279 | 0.150 | 0.200 | 0.214 | 0.169 | 0.337 | 0.174 | 0.213 | 0.055 | 0.171 | 0.279 | 0.274 |
| Vietnamese | Khmer | 0.079 | 0.006 | 0.052 | 0.099 | 0.093 | 0.018 | 0.043 | 0.005 | 0.003 | 0.365 | 0.013 | 0.065 | 0.003 | 0.004 | 0.096 | 0.078 |
| Vietnamese | Lao | 0.167 | 0.014 | 0.137 | 0.008 | 0.211 | 0.014 | 0.054 | 0.026 | 0.005 | 0.415 | 0.007 | 0.067 | 0.005 | 0.007 | 0.155 | 0.179 |
| Vietnamese | Burmese | 0.112 | 0.003 | 0.141 | 0.130 | 0.165 | 0.014 | 0.050 | 0.099 | 0.002 | 0.334 | 0.015 | 0.051 | 0.004 | 0.002 | 0.110 | 0.074 |
| Vietnamese | Filipino | 0.261 | 0.024 | 0.207 | 0.187 | 0.251 | 0.075 | 0.147 | 0.198 | 0.092 | 0.345 | 0.171 | 0.188 | 0.040 | 0.113 | 0.249 | 0.244 |
| Vietnamese | Indonesian | 0.330 | 0.312 | 0.304 | 0.231 | 0.298 | 0.224 | 0.269 | 0.294 | 0.304 | 0.353 | 0.226 | 0.253 | 0.084 | 0.236 | 0.345 | 0.316 |
| Vietnamese | English | 0.376 | 0.376 | 0.364 | 0.290 | 0.365 | 0.334 | 0.345 | 0.345 | 0.353 | 0.242 | 0.237 | 0.326 | 0.226 | 0.352 | 0.421 | 0.385 |
| Vietnamese | Chinese | 0.273 | 0.218 | 0.283 | 0.247 | 0.252 | 0.235 | 0.250 | 0.244 | 0.242 | 0.290 | 0.189 | 0.183 | 0.105 | 0.253 | 0.316 | 0.164 |
| English | Thai | 0.341 | 0.348 | 0.345 | 0.286 | 0.320 | 0.200 | 0.272 | 0.308 | 0.074 | 0.304 | 0.235 | 0.236 | 0.034 | 0.184 | 0.376 | 0.238 |
| English | Malay | 0.403 | 0.433 | 0.381 | 0.305 | 0.411 | 0.239 | 0.308 | 0.353 | 0.243 | 0.017 | 0.332 | 0.322 | 0.093 | 0.250 | 0.394 | 0.422 |
| English | Khmer | 0.091 | 0.007 | 0.060 | 0.115 | 0.125 | 0.022 | 0.051 | 0.008 | 0.004 | 0.271 | 0.013 | 0.078 | 0.003 | 0.004 | 0.122 | 0.090 |
| English | Lao | 0.205 | 0.017 | 0.172 | 0.010 | 0.274 | 0.017 | 0.075 | 0.034 | 0.006 | 0.242 | 0.006 | 0.074 | 0.005 | 0.007 | 0.197 | 0.232 |
| English | Burmese | 0.129 | 0.003 | 0.166 | 0.153 | 0.208 | 0.019 | 0.066 | 0.126 | 0.002 | 0.183 | 0.020 | 0.062 | 0.004 | 0.002 | 0.128 | 0.122 |
| English | Filipino | 0.354 | 0.031 | 0.297 | 0.250 | 0.352 | 0.120 | 0.220 | 0.294 | 0.132 | 0.350 | 0.272 | 0.270 | 0.069 | 0.177 | 0.332 | 0.362 |
| English | Indonesian | 0.478 | 0.468 | 0.470 | 0.321 | 0.448 | 0.328 | 0.401 | 0.463 | 0.439 | 0.017 | 0.410 | 0.364 | 0.134 | 0.362 | 0.499 | 0.467 |
| English | Vietnamese | 0.421 | 0.455 | 0.414 | 0.339 | 0.415 | 0.316 | 0.380 | 0.416 | 0.406 | 0.017 | 0.377 | 0.350 | 0.108 | 0.311 | 0.457 | 0.416 |
| English | Chinese | 0.341 | 0.271 | 0.355 | 0.327 | 0.329 | 0.307 | 0.328 | 0.316 | 0.304 | 0.017 | 0.271 | 0.242 | 0.183 | 0.350 | 0.396 | 0.205 |
| Chinese | Thai | 0.263 | 0.252 | 0.246 | 0.220 | 0.237 | 0.142 | 0.209 | 0.222 | 0.053 | 0.017 | 0.166 | 0.175 | 0.023 | 0.129 | 0.283 | 0.171 |
| Chinese | Malay | 0.221 | 0.234 | 0.200 | 0.186 | 0.223 | 0.119 | 0.164 | 0.181 | 0.132 | 0.017 | 0.156 | 0.162 | 0.035 | 0.138 | 0.245 | 0.218 |
| Chinese | Khmer | 0.073 | 0.006 | 0.062 | 0.090 | 0.087 | 0.018 | 0.036 | 0.005 | 0.004 | 0.017 | 0.010 | 0.063 | 0.002 | 0.005 | 0.085 | 0.065 |
| Chinese | Lao | 0.135 | 0.011 | 0.112 | 0.007 | 0.174 | 0.010 | 0.043 | 0.020 | 0.004 | 0.017 | 0.004 | 0.042 | 0.003 | 0.005 | 0.135 | 0.138 |
| Chinese | Burmese | 0.097 | 0.002 | 0.133 | 0.122 | 0.164 | 0.014 | 0.055 | 0.101 | 0.001 | 0.017 | 0.016 | 0.048 | 0.002 | 0.002 | 0.107 | 0.114 |
| Chinese | Filipino | 0.212 | 0.019 | 0.171 | 0.160 | 0.198 | 0.058 | 0.120 | 0.160 | 0.067 | 0.017 | 0.134 | 0.146 | 0.029 | 0.088 | 0.214 | 0.189 |
| Chinese | Indonesian | 0.269 | 0.264 | 0.253 | 0.207 | 0.248 | 0.179 | 0.224 | 0.236 | 0.255 | 0.017 | 0.203 | 0.187 | 0.054 | 0.197 | 0.299 | 0.241 |
| Chinese | Vietnamese | 0.306 | 0.319 | 0.289 | 0.258 | 0.292 | 0.181 | 0.274 | 0.276 | 0.288 | 0.017 | 0.248 | 0.247 | 0.060 | 0.222 | 0.334 | 0.292 |
| Chinese | English | 0.301 | 0.282 | 0.292 | 0.256 | 0.292 | 0.273 | 0.288 | 0.275 | 0.290 | 0.017 | 0.271 | 0.255 | 0.187 | 0.306 | 0.347 | 0.304 |

## A.7 Detailed COMET Breakdown

Table 6 reports COMET scores for every evaluated source–target direction across MtPO and all baselines, complementing the BLEU breakdown in Appendix A.6.

**Table 6:** COMET scores per source–target language pair.

| Source | Target | Model COMET Scores | | | | | | | | | | | | | | | |
|---|---|---|---|---|---|---|---|---|---|---|---|---|---|---|---|---|---|---|
| | | Gem3 | SeedX | QwenMT | Huny | MtPO | Qw2.5 | Qw3 | Apert | Aya | Emma | LLa3.1 | LLaX3 | Mistr | Tower | Google | NLLB |
| Thai | Malay | 0.8774 | 0.876 | 0.877 | 0.8729 | 0.8741 | 0.5088 | 0.8537 | 0.8627 | 0.7755 | 0.4776 | 0.7004 | 0.8471 | 0.5788 | 0.7917 | 0.8481 | 0.8572 |
| Thai | Khmer | 0.7952 | 0.4917 | 0.7807 | 0.8126 | 0.8245 | 0.3895 | 0.6391 | 0.5416 | 0.3694 | 0.7217 | 0.5474 | 0.6985 | 0.4164 | 0.3355 | 0.7682 | 0.7884 |
| Thai | Lao | 0.8223 | 0.51 | 0.7957 | 0.6894 | 0.8409 | 0.3552 | 0.6412 | 0.5743 | 0.3659 | 0.5655 | 0.5892 | 0.6166 | 0.3353 | 0.4796 | 0.7615 | 0.8295 |
| Thai | Burmese | 0.8373 | 0.4567 | 0.8487 | 0.8498 | 0.8673 | 0.4155 | 0.6412 | 0.5807 | 0.3781 | 0.7714 | 0.5436 | 0.6405 | 0.388 | 0.3276 | 0.7535 | 0.8205 |
| Thai | Filipino | 0.8401 | 0.655 | 0.8298 | 0.8244 | 0.8353 | 0.4611 | 0.7514 | 0.8136 | 0.5903 | 0.5316 | 0.7156 | 0.7974 | 0.5538 | 0.6924 | 0.749 | 0.819 |
| Thai | Indonesian | 0.901 | 0.8988 | 0.8988 | 0.8907 | 0.8898 | 0.5243 | 0.8843 | 0.889 | 0.8209 | 0.4651 | 0.737 | 0.8678 | 0.6252 | 0.8546 | 0.8923 | 0.878 |
| Thai | Vietnamese | 0.8892 | 0.8867 | 0.8896 | 0.883 | 0.8808 | 0.504 | 0.8762 | 0.8739 | 0.8117 | 0.4553 | 0.7257 | 0.8615 | 0.5719 | 0.8324 | 0.8776 | 0.8685 |
| Thai | English | 0.8902 | 0.8863 | 0.8899 | 0.8782 | 0.8847 | 0.5233 | 0.8813 | 0.87 | 0.8069 | 0.685 | 0.6816 | 0.8731 | 0.7952 | 0.8778 | 0.8804 | 0.8743 |
| Thai | Chinese | 0.8794 | 0.8745 | 0.8841 | 0.8778 | 0.8736 | 0.5127 | 0.8727 | 0.8684 | 0.7914 | 0.6146 | 0.8102 | 0.8473 | 0.7292 | 0.8671 | 0.873 | 0.819 |
| Malay | Thai | 0.8692 | 0.8667 | 0.8694 | 0.8674 | 0.8553 | 0.4569 | 0.8367 | 0.8373 | 0.5409 | 0.5551 | 0.8041 | 0.8153 | 0.4664 | 0.7574 | 0.864 | 0.8242 |
| Malay | Khmer | 0.7774 | 0.4523 | 0.759 | 0.8048 | 0.8129 | 0.3786 | 0.6218 | 0.5215 | 0.3672 | 0.6755 | 0.5314 | 0.6893 | 0.3848 | 0.3524 | 0.7558 | 0.7828 |
| Malay | Lao | 0.8097 | 0.4699 | 0.7704 | 0.6461 | 0.8361 | 0.3476 | 0.5987 | 0.5413 | 0.3611 | 0.5203 | 0.3883 | 0.594 | 0.318 | 0.3955 | 0.7544 | 0.818 |
| Malay | Burmese | 0.8333 | 0.4167 | 0.8383 | 0.849 | 0.8636 | 0.4207 | 0.6512 | 0.7796 | 0.3867 | 0.7784 | 0.542 | 0.6458 | 0.3925 | 0.3405 | 0.7315 | 0.828 |
| Malay | Filipino | 0.8501 | 0.6247 | 0.8327 | 0.8404 | 0.8459 | 0.4362 | 0.7447 | 0.7946 | 0.6319 | 0.59 | 0.7674 | 0.8042 | 0.5606 | 0.6912 | 0.7353 | 0.8367 |
| Malay | Indonesian | 0.9181 | 0.9182 | 0.9149 | 0.9138 | 0.9038 | 0.5051 | 0.9016 | 0.895 | 0.9023 | 0.6664 | 0.8935 | 0.8869 | 0.7003 | 0.8818 | 0.915 | 0.9072 |
| Malay | Vietnamese | 0.8832 | 0.8841 | 0.8817 | 0.8806 | 0.8702 | 0.4708 | 0.8578 | 0.8556 | 0.8606 | 0.5954 | 0.8365 | 0.8432 | 0.5596 | 0.8107 | 0.8834 | 0.8639 |
| Malay | English | 0.894 | 0.8932 | 0.893 | 0.8842 | 0.8918 | 0.4882 | 0.8794 | 0.8822 | 0.8715 | 0.6742 | 0.866 | 0.8804 | 0.8407 | 0.8778 | 0.8978 | 0.8907 |
| Malay | Chinese | 0.8639 | 0.8635 | 0.8697 | 0.8708 | 0.8589 | 0.4868 | 0.8545 | 0.8447 | 0.8357 | 0.6041 | 0.8305 | 0.8276 | 0.7354 | 0.8468 | 0.8711 | 0.8098 |
| Khmer | Thai | 0.8581 | 0.6512 | 0.8556 | 0.8462 | 0.8522 | 0.4609 | 0.7295 | 0.7711 | 0.4606 | 0.6524 | 0.6795 | 0.8166 | 0.4325 | 0.6588 | 0.8496 | 0.8123 |
| Khmer | Malay | 0.8606 | 0.6687 | 0.857 | 0.846 | 0.8588 | 0.4826 | 0.8141 | 0.8376 | 0.5707 | 0.5084 | 0.4516 | 0.8246 | 0.513 | 0.6724 | 0.8368 | 0.8453 |
| Khmer | Lao | 0.8063 | 0.4383 | 0.7616 | 0.6677 | 0.8396 | 0.4369 | 0.6654 | 0.4637 | 0.376 | 0.6479 | 0.4812 | 0.6119 | 0.4404 | 0.4293 | 0.7544 | 0.8159 |
| Khmer | Burmese | 0.81 | 0.4041 | 0.8207 | 0.8413 | 0.8579 | 0.4117 | 0.591 | 0.4325 | 0.392 | 0.604 | 0.5353 | 0.63 | 0.3885 | 0.3217 | 0.7387 | 0.8233 |
| Khmer | Filipino | 0.8345 | 0.5475 | 0.8213 | 0.8137 | 0.8305 | 0.4754 | 0.7228 | 0.8023 | 0.5273 | 0.5005 | 0.4547 | 0.7842 | 0.5054 | 0.619 | 0.817 | 0.8128 |
| Khmer | Indonesian | 0.8811 | 0.686 | 0.8747 | 0.8624 | 0.8743 | 0.4976 | 0.8353 | 0.8592 | 0.599 | 0.5223 | 0.4594 | 0.8408 | 0.5387 | 0.7165 | 0.8754 | 0.8613 |
| Khmer | Vietnamese | 0.876 | 0.6836 | 0.8739 | 0.8591 | 0.8698 | 0.485 | 0.8359 | 0.853 | 0.5866 | 0.5455 | 0.4726 | 0.8424 | 0.4999 | 0.706 | 0.8678 | 0.8524 |
| Khmer | English | 0.8748 | 0.6759 | 0.8714 | 0.8545 | 0.8763 | 0.5095 | 0.8424 | 0.8635 | 0.5871 | 0.4808 | 0.6344 | 0.8564 | 0.6086 | 0.7543 | 0.8669 | 0.8632 |
| Khmer | Chinese | 0.8582 | 0.6376 | 0.8584 | 0.8474 | 0.8548 | 0.4873 | 0.825 | 0.8387 | 0.5706 | 0.5558 | 0.53 | 0.8174 | 0.5608 | 0.739 | 0.8543 | 0.8093 |
| Lao | Thai | 0.8686 | 0.5347 | 0.8645 | 0.5859 | 0.8593 | 0.4395 | 0.7585 | 0.6251 | 0.4541 | 0.592 | 0.6764 | 0.8107 | 0.4399 | 0.6397 | 0.8663 | 0.8308 |
| Lao | Malay | 0.8685 | 0.5192 | 0.8598 | 0.5998 | 0.867 | 0.4651 | 0.8148 | 0.8345 | 0.5581 | 0.4646 | 0.4544 | 0.8046 | 0.4622 | 0.6311 | 0.8527 | 0.853 |

*Continued on next page*

| Source | Target | Model COMET Scores | | | | | | | | | | | | | | | |
|---|---|---|---|---|---|---|---|---|---|---|---|---|---|---|---|---|---|
| | | Gem3 | SeedX | QwenMT | Huny | MtPO | Qw2.5 | Qw3 | Apert | Aya | Emma | LLa3.1 | LLaX3 | Mistr | Tower | Google | NLLB |
| Lao | Khmer | 0.7796 | 0.357 | 0.7379 | 0.5851 | 0.8223 | 0.4228 | 0.6395 | 0.4829 | 0.3845 | 0.6404 | 0.5064 | 0.6895 | 0.4269 | 0.3454 | 0.7699 | 0.7867 |
| Lao | Burmese | 0.8237 | 0.3396 | 0.7731 | 0.6355 | 0.8592 | 0.3896 | 0.5821 | 0.4228 | 0.3871 | 0.7017 | 0.5037 | 0.6057 | 0.3894 | 0.3252 | 0.7462 | 0.821 |
| Lao | Filipino | 0.8403 | 0.4352 | 0.8244 | 0.6208 | 0.8343 | 0.4693 | 0.7274 | 0.7944 | 0.5273 | 0.4442 | 0.5359 | 0.7672 | 0.4763 | 0.5864 | 0.8034 | 0.8186 |
| Lao | Indonesian | 0.8862 | 0.5362 | 0.8772 | 0.608 | 0.8785 | 0.4753 | 0.8358 | 0.8525 | 0.5865 | 0.4798 | 0.5206 | 0.817 | 0.4876 | 0.6709 | 0.8897 | 0.8663 |
| Lao | Vietnamese | 0.8777 | 0.5401 | 0.8708 | 0.5909 | 0.8698 | 0.4635 | 0.8301 | 0.8424 | 0.571 | 0.4911 | 0.5183 | 0.8188 | 0.4642 | 0.6514 | 0.8786 | 0.8557 |
| Lao | English | 0.8793 | 0.5099 | 0.8737 | 0.5908 | 0.8807 | 0.4873 | 0.8407 | 0.8553 | 0.5803 | 0.5519 | 0.6795 | 0.8317 | 0.5094 | 0.6973 | 0.8858 | 0.8703 |
| Lao | Chinese | 0.8613 | 0.5356 | 0.8574 | 0.5858 | 0.8576 | 0.4659 | 0.8196 | 0.8275 | 0.5594 | 0.5716 | 0.5444 | 0.7892 | 0.512 | 0.6763 | 0.8632 | 0.8073 |
| Burmese | Thai | 0.8511 | 0.5809 | 0.8523 | 0.8312 | 0.841 | 0.4595 | 0.6714 | 0.7998 | 0.43 | 0.6321 | 0.6905 | 0.7883 | 0.4255 | 0.6075 | 0.8452 | 0.8112 |
| Burmese | Malay | 0.8529 | 0.5939 | 0.8516 | 0.8359 | 0.8425 | 0.4844 | 0.7907 | 0.8261 | 0.5439 | 0.5019 | 0.4221 | 0.7981 | 0.4912 | 0.6526 | 0.8337 | 0.854 |
| Burmese | Khmer | 0.7468 | 0.4073 | 0.7621 | 0.786 | 0.8054 | 0.4037 | 0.5781 | 0.5215 | 0.3822 | 0.6282 | 0.5061 | 0.6611 | 0.4895 | 0.3253 | 0.753 | 0.7675 |
| Burmese | Lao | 0.7794 | 0.4128 | 0.7622 | 0.6343 | 0.8204 | 0.4153 | 0.5798 | 0.5099 | 0.3728 | 0.5964 | 0.38 | 0.5671 | 0.2983 | 0.3732 | 0.7487 | 0.8024 |
| Burmese | Filipino | 0.8318 | 0.513 | 0.8213 | 0.8111 | 0.8257 | 0.4853 | 0.7015 | 0.7943 | 0.5176 | 0.5707 | 0.7573 | 0.5029 | 0.5997 | | 0.805 | 0.8233 |
| Burmese | Indonesian | 0.8727 | 0.6055 | 0.8715 | 0.8512 | 0.8636 | 0.4964 | 0.8172 | 0.847 | 0.5667 | 0.527 | 0.4432 | 0.8176 | 0.5192 | 0.6989 | 0.8724 | 0.8694 |
| Burmese | Vietnamese | 0.8649 | 0.5947 | 0.8647 | 0.8437 | 0.8558 | 0.4813 | 0.8058 | 0.8341 | 0.5538 | 0.5142 | 0.4647 | 0.8222 | 0.4872 | 0.6751 | 0.8621 | 0.8582 |
| Burmese | English | 0.8687 | 0.6009 | 0.871 | 0.8483 | 0.869 | 0.5134 | 0.8284 | 0.8573 | 0.5546 | 0.4874 | 0.7366 | 0.8461 | 0.5704 | 0.7379 | 0.8723 | 0.8729 |
| Burmese | Chinese | 0.8519 | 0.5612 | 0.8541 | 0.835 | 0.8433 | 0.4791 | 0.8073 | 0.8281 | 0.5319 | 0.5455 | 0.631 | 0.8085 | 0.5252 | 0.7137 | 0.8519 | 0.8147 |
| Filipino | Thai | 0.8605 | 0.8382 | 0.8561 | 0.8524 | 0.8456 | 0.4445 | 0.8054 | 0.8014 | 0.5221 | 0.5238 | 0.7758 | 0.8047 | 0.4483 | 0.7238 | 0.8522 | 0.8155 |
| Filipino | Malay | 0.8777 | 0.8614 | 0.8617 | 0.8691 | 0.8722 | 0.458 | 0.7999 | 0.8118 | 0.7906 | 0.5144 | 0.8009 | 0.837 | 0.5611 | 0.7679 | 0.8551 | 0.8657 |
| Filipino | Khmer | 0.7747 | 0.4293 | 0.7503 | 0.7947 | 0.8045 | 0.3853 | 0.5913 | 0.4993 | 0.3611 | 0.5651 | 0.5112 | 0.6668 | 0.3642 | 0.3355 | 0.748 | 0.776 |
| Filipino | Lao | 0.8021 | 0.4473 | 0.7667 | 0.6351 | 0.8275 | 0.3457 | 0.5616 | 0.5115 | 0.3566 | 0.4764 | 0.3677 | 0.5847 | 0.2998 | 0.3739 | 0.7417 | 0.8063 |
| Filipino | Burmese | 0.8343 | 0.3965 | 0.8318 | 0.8455 | 0.8591 | 0.4183 | 0.6254 | 0.7175 | 0.3772 | 0.758 | 0.5324 | 0.6382 | 0.3842 | 0.3343 | 0.7275 | 0.8202 |
| Filipino | Indonesian | 0.8988 | 0.8808 | 0.8832 | 0.8865 | 0.8871 | 0.4716 | 0.8227 | 0.8358 | 0.8294 | 0.5367 | 0.8238 | 0.8565 | 0.6171 | 0.8012 | 0.9005 | 0.8816 |
| Filipino | Vietnamese | 0.872 | 0.8498 | 0.8662 | 0.86 | 0.8572 | 0.4492 | 0.8184 | 0.8266 | 0.8044 | 0.4842 | 0.8029 | 0.8262 | 0.528 | 0.7597 | 0.8697 | 0.849 |
| Filipino | English | 0.8841 | 0.8599 | 0.8753 | 0.8659 | 0.878 | 0.4702 | 0.8509 | 0.8583 | 0.8147 | 0.5923 | 0.8516 | 0.8613 | 0.792 | 0.8423 | 0.8888 | 0.8752 |
| Filipino | Chinese | 0.8558 | 0.8225 | 0.8552 | 0.8536 | 0.8453 | 0.4706 | 0.8271 | 0.8231 | 0.7776 | 0.532 | 0.7996 | 0.809 | 0.6868 | 0.8132 | 0.8609 | 0.7949 |
| Indonesian | Thai | 0.8749 | 0.8739 | 0.8777 | 0.8738 | 0.8644 | 0.4638 | 0.8522 | 0.8514 | 0.5462 | 0.5688 | 0.8158 | 0.825 | 0.4658 | 0.7675 | 0.8674 | 0.8328 |
| Indonesian | Malay | 0.9045 | 0.9063 | 0.9055 | 0.903 | 0.9035 | 0.5139 | 0.8968 | 0.897 | 0.8722 | 0.6736 | 0.8954 | 0.8838 | 0.6599 | 0.8671 | 0.8872 | 0.9005 |
| Indonesian | Khmer | 0.7823 | 0.4582 | 0.7613 | 0.8062 | 0.8151 | 0.382 | 0.6241 | 0.5282 | 0.365 | 0.6913 | 0.5312 | 0.6892 | 0.3886 | 0.3423 | 0.7537 | 0.7866 |
| Indonesian | Lao | 0.8096 | 0.4755 | 0.7655 | 0.6464 | 0.8374 | 0.3468 | 0.5954 | 0.55 | 0.3618 | 0.5377 | 0.3846 | 0.589 | 0.3173 | 0.3909 | 0.7447 | 0.8172 |
| Indonesian | Burmese | 0.8313 | 0.4219 | 0.8374 | 0.8508 | 0.865 | 0.4237 | 0.652 | 0.7858 | 0.3862 | 0.7847 | 0.5436 | 0.653 | 0.3897 | 0.337 | 0.7263 | 0.8284 |
| Indonesian | Filipino | 0.853 | 0.6329 | 0.8365 | 0.8409 | 0.848 | 0.4428 | 0.7503 | 0.8096 | 0.6463 | 0.6079 | 0.7784 | 0.8132 | 0.5626 | 0.6966 | 0.8227 | 0.8412 |
| Indonesian | Vietnamese | 0.8903 | 0.8889 | 0.8913 | 0.8875 | 0.8795 | 0.4782 | 0.8714 | 0.8732 | 0.8814 | 0.6108 | 0.8558 | 0.8539 | 0.5656 | 0.8315 | 0.8998 | 0.8856 |
| Indonesian | English | 0.8997 | 0.8989 | 0.9002 | 0.8895 | 0.8966 | 0.4962 | 0.8912 | 0.8932 | 0.893 | 0.6712 | 0.8736 | 0.8852 | 0.8473 | 0.8896 | 0.8943 | 0.8738 |
| Indonesian | Chinese | 0.8736 | 0.8725 | 0.8797 | 0.8765 | 0.8665 | 0.4948 | 0.8672 | 0.86 | 0.8607 | 0.6167 | 0.8441 | 0.841 | 0.7497 | 0.8632 | 0.8688 | 0.8263 |
| Vietnamese | Thai | 0.8737 | 0.8633 | 0.8771 | 0.8708 | 0.8643 | 0.4677 | 0.8532 | 0.8519 | 0.5504 | 0.5145 | 0.8187 | 0.8313 | 0.4651 | 0.7701 | 0.8877 | 0.8426 |
| Vietnamese | Malay | 0.8756 | 0.8758 | 0.8744 | 0.8729 | 0.8731 | 0.4887 | 0.8526 | 0.8542 | 0.8467 | 0.4586 | 0.8305 | 0.8506 | 0.6019 | 0.7902 | 0.8673 | 0.814 |
| Vietnamese | Khmer | 0.7882 | 0.4521 | 0.7735 | 0.8122 | 0.8251 | 0.3805 | 0.6416 | 0.5395 | 0.37 | 0.7 | 0.5408 | 0.6845 | 0.3931 | 0.344 | 0.7593 | 0.7669 |
| Vietnamese | Lao | 0.8138 | 0.4696 | 0.7771 | 0.654 | 0.8418 | 0.3454 | 0.6021 | 0.5602 | 0.3608 | 0.5385 | 0.3875 | 0.6026 | 0.3374 | 0.3908 | 0.7413 | 0.7968 |
| Vietnamese | Burmese | 0.8338 | 0.4176 | 0.8424 | 0.8484 | 0.8636 | 0.4213 | 0.6532 | 0.7743 | 0.3811 | 0.7603 | 0.5443 | 0.6409 | 0.3883 | 0.33 | 0.7321 | 0.8241 |
| Vietnamese | Filipino | 0.8316 | 0.621 | 0.8129 | 0.8161 | 0.8256 | 0.4394 | 0.7284 | 0.7966 | 0.6192 | 0.5686 | 0.7565 | 0.7892 | 0.54 | 0.6711 | 0.7917 | 0.8034 |
| Vietnamese | Indonesian | 0.9005 | 0.8999 | 0.8985 | 0.894 | 0.8918 | 0.5098 | 0.8854 | 0.8887 | 0.8934 | 0.5278 | 0.8641 | 0.8754 | 0.6656 | 0.8556 | 0.8943 | 0.8738 |
| Vietnamese | English | 0.8812 | 0.8817 | 0.8813 | 0.8729 | 0.8776 | 0.4936 | 0.8745 | 0.8746 | 0.8754 | 0.6089 | 0.8307 | 0.8676 | 0.8268 | 0.8725 | 0.8881 | 0.8725 |
| Vietnamese | Chinese | 0.8774 | 0.8781 | 0.8821 | 0.88 | 0.8709 | 0.5051 | 0.8729 | 0.8669 | 0.8691 | 0.6133 | 0.8495 | 0.8485 | 0.7636 | 0.869 | 0.882 | 0.8654 |
| English | Thai | 0.8951 | 0.8937 | 0.8979 | 0.8935 | 0.887 | 0.4773 | 0.8748 | 0.8746 | 0.5566 | 0.6488 | 0.8488 | 0.8462 | 0.4679 | 0.7985 | 0.8673 | 0.8140 |
| English | Malay | 0.9022 | 0.9051 | 0.9027 | 0.9015 | 0.9018 | 0.4878 | 0.8802 | 0.8728 | 0.7139 | | 0.8822 | 0.8695 | 0.6266 | 0.807 | 0.8533 | |
| English | Khmer | 0.8107 | 0.4166 | 0.7876 | 0.8359 | 0.8473 | 0.3943 | 0.6634 | 0.5549 | 0.3766 | 0.771 | 0.557 | 0.7119 | 0.4127 | 0.3625 | 0.7593 | 0.7669 |
| English | Lao | 0.8339 | 0.4387 | 0.7963 | 0.6894 | 0.8657 | 0.3527 | 0.637 | 0.5758 | 0.3669 | 0.6124 | 0.4244 | 0.6003 | 0.3552 | 0.4038 | 0.7413 | 0.7968 |
| English | Burmese | 0.862 | 0.3756 | 0.8692 | 0.8779 | 0.8928 | 0.4439 | 0.6922 | 0.8214 | 0.398 | 0.8273 | 0.5743 | 0.6986 | 0.4256 | 0.3551 | 0.7321 | 0.8241 |
| English | Filipino | 0.8611 | 0.6245 | 0.8431 | 0.8478 | 0.8526 | 0.4375 | 0.7625 | 0.8268 | 0.6438 | 0.5863 | 0.8007 | 0.8105 | 0.5618 | 0.7025 | 0.7917 | 0.8034 |
| English | Indonesian | 0.9254 | 0.9236 | 0.9256 | 0.9192 | 0.9152 | 0.5106 | 0.9092 | 0.9139 | 0.9172 | 0.6491 | 0.9056 | 0.8883 | 0.6879 | 0.8826 | 0.8943 | 0.8738 |
| English | Vietnamese | 0.9008 | 0.903 | 0.903 | 0.9 | 0.8912 | 0.4867 | 0.8895 | 0.8886 | 0.8942 | 0.6617 | 0.8785 | 0.8631 | 0.588 | 0.841 | 0.8881 | 0.8725 |
| English | Chinese | 0.891 | 0.8921 | 0.897 | 0.8987 | 0.8867 | 0.5055 | 0.8877 | 0.8772 | 0.88 | 0.6424 | 0.8713 | 0.8568 | 0.8237 | 0.8906 | 0.882 | 0.8654 |
| Chinese | Thai | 0.8748 | 0.8723 | 0.8753 | 0.8743 | 0.8671 | 0.4729 | 0.8543 | 0.8545 | 0.5446 | 0.5942 | 0.8168 | 0.8283 | 0.4602 | 0.773 | 0.8751 | 0.8610 |
| Chinese | Malay | 0.8679 | 0.8704 | 0.8673 | 0.8688 | 0.8687 | 0.4923 | 0.8496 | 0.8558 | 0.8394 | 0.5408 | 0.841 | 0.8381 | 0.5623 | 0.7818 | 0.8709 | 0.8722 |
| Chinese | Khmer | 0.787 | 0.4623 | 0.7701 | 0.8078 | 0.8251 | 0.3858 | 0.6328 | 0.5361 | 0.3679 | 0.7537 | 0.5263 | 0.6647 | 0.388 | 0.3429 | 0.8751 | 0.8731 |
| Chinese | Lao | 0.8054 | 0.4784 | 0.7673 | 0.6488 | 0.8351 | 0.3485 | 0.5973 | 0.5417 | 0.3599 | 0.591 | 0.3756 | 0.5629 | 0.3116 | 0.3719 | 0.8787 | 0.8749 |
| Chinese | Burmese | 0.8293 | 0.4281 | 0.847 | 0.8489 | 0.8667 | 0.4207 | 0.6734 | 0.7982 | 0.3822 | 0.7868 | 0.543 | 0.6467 | 0.3966 | 0.3376 | 0.8779 | 0.8745 |
| Chinese | Filipino | 0.8235 | 0.6283 | 0.8105 | 0.8142 | 0.8196 | 0.4451 | 0.7311 | 0.7953 | 0.6251 | 0.5728 | 0.762 | 0.7821 | 0.5481 | 0.6699 | 0.8709 | 0.8722 |
| Chinese | Indonesian | 0.8918 | 0.8928 | 0.8933 | 0.8899 | 0.8862 | 0.5143 | 0.8821 | 0.883 | 0.8859 | 0.569 | 0.8731 | 0.8626 | 0.6312 | 0.8519 | 0.8584 | 0.8660 |
| Chinese | Vietnamese | 0.8883 | 0.8883 | 0.8898 | 0.8878 | 0.8833 | 0.49 | 0.8801 | 0.879 | 0.8825 | 0.4904 | 0.8658 | 0.8585 | 0.5601 | 0.8391 | 0.861 | 0.8454 |
| Chinese | English | 0.8749 | 0.8745 | 0.8779 | 0.8731 | 0.8751 | 0.5027 | 0.8738 | 0.8709 | 0.8722 | 0.6584 | 0.866 | 0.861 | 0.8454 | 0.8772 | 0.8738 | 0.8772 |

## A.8 DETAILED chrF BREAKDOWN

Table 7 reports chrF scores for every evaluated source–target direction across MtPO and all baselines, complementing the BLEU and COMET breakdowns in Appendices A.6 and A.7.

**Table 7:** chrF scores per source–target language pair.

| Source | Target | Model chrF Scores | | | | | | | | | | | | | | | |
|---|---|---|---|---|---|---|---|---|---|---|---|---|---|---|---|---|---|
| | | Gem3 | SeedX | QwenMT | Huny | MtPO | Qw2.5 | Qw3 | Apert | Aya | Emma | LLa3.1 | LLaX3 | Mistr | Tower | Google | NLLB |
| Thai | Malay | 58.31 | 57.57 | 55.44 | 55.74 | 57.95 | 23.09 | 51.79 | 53.82 | 42.32 | 6.43 | 4.89 | 50.21 | 35.48 | 48.86 | 58.05 | 57.71 |
| Thai | Khmer | 36.93 | 2.85 | 34.34 | 39.40 | 42.28 | 13.25 | 28.35 | 15.82 | 11.60 | 35.03 | 18.45 | 30.47 | 1.76 | 11.74 | 41.98 | 41.85 |
| Thai | Lao | 45.85 | 4.08 | 41.23 | 2.48 | 48.60 | 11.28 | 31.41 | 18.32 | 10.51 | 2.03 | 4.79 | 27.54 | 3.21 | 9.20 | 46.82 | 48.23 |
| Thai | Burmese | 39.40 | 0.71 | 42.25 | 44.15 | 46.55 | 17.47 | 29.83 | 36.38 | 10.15 | 31.80 | 16.20 | 25.48 | 15.08 | 11.84 | 44.23 | 44.18 |
| Thai | Filipino | 54.12 | 18.07 | 49.96 | 51.67 | 53.14 | 23.36 | 44.68 | 49.49 | 36.54 | 14.57 | 31.06 | 46.40 | 32.88 | 41.20 | 48.17 | 53.69 |
| Thai | Indonesian | 60.90 | 58.45 | 58.20 | 56.20 | 57.78 | 21.94 | 55.93 | 57.70 | 48.26 | 9.50 | 10.19 | 52.13 | 37.91 | 52.99 | 60.14 | 58.28 |
| Thai | Vietnamese | 52.90 | 53.38 | 51.56 | 49.98 | 51.14 | 14.68 | 48.79 | 50.00 | 41.03 | 9.84 | 5.96 | 46.57 | 28.68 | 44.09 | 54.21 | 52.74 |
| Thai | English | 61.02 | 60.64 | 60.07 | 57.14 | 59.96 | 19.80 | 58.72 | 58.80 | 46.20 | 23.22 | 9.04 | 55.28 | 44.48 | 58.76 | 61.19 | 60.53 |
| Thai | Chinese | 31.87 | 27.60 | 31.70 | 30.94 | 31.01 | 3.20 | 29.94 | 29.69 | 20.08 | 8.31 | 15.89 | 24.94 | 15.53 | 29.27 | 31.84 | 28.46 |
| Malay | Thai | 53.36 | 52.86 | 52.85 | 52.08 | 51.18 | 13.34 | 47.37 | 48.52 | 26.65 | 11.77 | 43.31 | 44.29 | 23.83 | 40.03 | 57.23 | 50.36 |
| Malay | Khmer | 36.97 | 3.09 | 34.63 | 40.44 | 43.76 | 12.33 | 28.31 | 16.32 | 11.94 | 31.98 | 18.65 | 31.46 | 2.42 | 12.51 | 44.05 | 43.62 |
| Malay | Lao | 45.17 | 4.43 | 40.82 | 3.34 | 50.31 | 10.65 | 28.23 | 17.24 | 11.31 | 3.19 | 10.82 | 26.76 | 4.77 | 10.64 | 49.35 | 50.17 |
| Malay | Burmese | 39.99 | 0.74 | 43.32 | 44.90 | 47.39 | 17.23 | 30.36 | 36.98 | 9.68 | 35.66 | 16.96 | 26.26 | 15.65 | 11.78 | 44.11 | 46.74 |
| Malay | Filipino | 55.42 | 18.43 | 52.66 | 54.52 | 56.27 | 21.98 | 45.69 | 49.79 | 39.83 | 21.55 | 47.66 | 50.01 | 36.99 | 42.75 | 48.36 | 57.09 |
| Malay | Indonesian | 65.20 | 65.01 | 63.80 | 61.35 | 60.58 | 21.69 | 59.59 | 56.78 | 61.51 | 25.02 | 56.17 | 56.29 | 48.39 | 55.05 | 63.15 | 62.29 |
| Malay | Vietnamese | 55.33 | 56.22 | 54.47 | 52.57 | 53.08 | 14.26 | 50.20 | 51.44 | 50.92 | 16.92 | 48.43 | 49.16 | 32.45 | 45.34 | 59.37 | 57.40 |
| Malay | English | 68.09 | 68.67 | 67.91 | 63.93 | 68.02 | 19.21 | 64.47 | 66.24 | 62.65 | 25.37 | 59.14 | 64.38 | 57.52 | 65.84 | 73.87 | 70.94 |
| Malay | Chinese | 33.12 | 29.00 | 31.53 | 34.08 | 32.77 | 3.04 | 31.07 | 30.44 | 28.42 | 7.76 | 27.53 | 26.77 | 19.15 | 30.84 | 35.90 | 28.89 |
| Khmer | Thai | 50.66 | 26.66 | 49.50 | 47.13 | 49.32 | 14.24 | 21.20 | 39.45 | 17.69 | 1.22 | 23.59 | 42.75 | 19.35 | 31.55 | 52.70 | 48.06 |
| Khmer | Malay | 57.18 | 34.42 | 54.93 | 52.70 | 56.98 | 22.29 | 48.81 | 51.73 | 29.39 | 3.02 | 1.86 | 48.78 | 28.70 | 38.75 | 57.92 | 57.29 |
| Khmer | Lao | 42.79 | 3.45 | 35.02 | 3.11 | 49.58 | 0.30 | 3.27 | 1.35 | 11.38 | 2.73 | 4.51 | 25.89 | 2.50 | 7.73 | 47.45 | 49.78 |
| Khmer | Burmese | 35.89 | 0.59 | 40.14 | 42.71 | 46.24 | 0.10 | 17.48 | 2.31 | 9.73 | 0.53 | 14.70 | 24.77 | 5.36 | 10.78 | 43.80 | 44.55 |
| Khmer | Filipino | 53.59 | 16.98 | 50.15 | 49.66 | 53.51 | 23.45 | 41.58 | 48.40 | 30.93 | 3.06 | 5.70 | 45.27 | 27.92 | 36.05 | 51.17 | 53.41 |
| Khmer | Indonesian | 59.00 | 34.15 | 56.89 | 52.88 | 57.64 | 22.09 | 51.10 | 55.20 | 29.48 | 3.07 | 2.44 | 49.52 | 29.73 | 41.20 | 60.16 | 58.68 |
| Khmer | Vietnamese | 51.15 | 27.95 | 50.08 | 46.26 | 50.61 | 14.58 | 43.98 | 47.69 | 22.36 | 3.32 | 2.25 | 44.83 | 22.08 | 33.82 | 54.71 | 53.76 |
| Khmer | English | 59.29 | 32.50 | 58.49 | 53.24 | 60.18 | 19.57 | 52.82 | 57.40 | 26.13 | 4.26 | 27.97 | 53.64 | 28.51 | 42.12 | 61.45 | 61.57 |
| Khmer | Chinese | 30.00 | 6.96 | 29.13 | 27.15 | 30.13 | 3.02 | 24.54 | 26.76 | 9.19 | 4.46 | 4.92 | 22.90 | 9.16 | 18.29 | 31.36 | 27.63 |
| Lao | Thai | 53.51 | 19.50 | 51.94 | 21.41 | 52.14 | 13.95 | 27.72 | 22.10 | 16.61 | 2.26 | 31.72 | 43.82 | 11.28 | 33.47 | 56.23 | 51.67 |
| Lao | Malay | 59.08 | 22.21 | 54.99 | 28.71 | 59.59 | 21.85 | 49.94 | 52.21 | 28.23 | 3.88 | 4.12 | 46.79 | 23.95 | 36.79 | 59.77 | 59.54 |
| Lao | Khmer | 36.54 | 2.06 | 28.59 | 20.48 | 44.29 | 1.35 | 4.05 | 4.33 | 11.75 | 32.23 | 13.79 | 31.01 | 1.89 | 9.41 | 43.58 | 44.48 |
| Lao | Burmese | 38.58 | 0.51 | 34.24 | 23.11 | 46.71 | 2.22 | 14.68 | 1.83 | 9.87 | 28.56 | 12.80 | 23.15 | 7.06 | 10.08 | 44.46 | 45.97 |
| Lao | Filipino | 54.40 | 13.62 | 49.73 | 30.03 | 53.79 | 22.26 | 42.94 | 47.37 | 30.29 | 4.31 | 20.57 | 43.26 | 24.82 | 33.13 | 54.08 | 55.44 |
| Lao | Indonesian | 60.41 | 22.94 | 57.05 | 28.00 | 58.86 | 21.38 | 51.30 | 54.34 | 28.56 | 5.33 | 13.34 | 46.90 | 25.10 | 38.26 | 63.01 | 60.97 |
| Lao | Vietnamese | 52.59 | 19.44 | 49.90 | 21.93 | 51.63 | 14.10 | 44.69 | 46.63 | 21.58 | 6.29 | 9.14 | 42.32 | 18.66 | 31.00 | 55.83 | 54.82 |
| Lao | English | 61.81 | 19.43 | 59.31 | 26.61 | 62.85 | 19.32 | 54.39 | 56.53 | 25.76 | 15.91 | 34.17 | 50.04 | 19.60 | 37.94 | 65.50 | 65.03 |
| Lao | Chinese | 30.81 | 3.73 | 29.58 | 10.58 | 30.84 | 2.76 | 24.66 | 25.99 | 8.66 | 9.57 | 6.24 | 20.34 | 6.67 | 15.40 | 33.00 | 28.06 |
| Burmese | Thai | 47.16 | 21.14 | 46.17 | 42.37 | 44.92 | 13.89 | 11.29 | 38.64 | 9.29 | 0.34 | 27.42 | 37.41 | 16.76 | 21.56 | 49.26 | 45.76 |
| Burmese | Malay | 53.29 | 27.57 | 50.86 | 49.19 | 50.46 | 21.48 | 42.35 | 47.57 | 26.60 | 1.49 | 1.13 | 42.24 | 24.73 | 34.03 | 52.57 | 54.14 |
| Burmese | Khmer | 30.46 | 1.24 | 32.70 | 34.89 | 38.73 | 9.58 | 13.20 | 13.03 | 9.35 | 11.91 | 15.19 | 26.75 | 0.58 | 8.38 | 41.06 | 40.91 |
| Burmese | Lao | 37.69 | 1.81 | 35.18 | 2.17 | 43.53 | 1.36 | 11.39 | 12.73 | 9.45 | 0.71 | 8.93 | 22.23 | 1.86 | 4.54 | 43.39 | 45.51 |
| Burmese | Filipino | 50.70 | 15.55 | 46.93 | 46.97 | 49.76 | 23.68 | 36.07 | 44.74 | 28.13 | 1.92 | 22.04 | 40.05 | 26.22 | 31.39 | 49.78 | 52.23 |
| Burmese | Indonesian | 54.89 | 27.10 | 53.13 | 49.02 | 52.79 | 21.34 | 44.79 | 50.34 | 26.33 | 1.88 | 2.67 | 43.39 | 25.99 | 36.04 | 57.36 | 57.19 |
| Burmese | Vietnamese | 47.50 | 20.18 | 46.06 | 42.42 | 46.05 | 14.01 | 36.95 | 43.05 | 18.94 | 2.79 | 6.41 | 39.11 | 18.46 | 28.17 | 51.63 | 51.12 |
| Burmese | English | 54.77 | 24.86 | 54.24 | 50.03 | 54.89 | 19.04 | 46.01 | 52.76 | 23.31 | 4.48 | 37.96 | 47.85 | 22.74 | 36.26 | 59.78 | 59.61 |
| Burmese | Chinese | 26.61 | 3.54 | 25.57 | 23.23 | 25.38 | 2.57 | 19.35 | 22.88 | 5.15 | 1.37 | 9.58 | 19.89 | 5.35 | 14.19 | 28.61 | 25.45 |
| Filipino | Thai | 53.40 | 49.08 | 51.53 | 50.86 | 50.61 | 13.56 | 45.07 | 44.51 | 24.99 | 5.19 | 41.13 | 43.46 | 22.99 | 38.66 | 55.72 | 50.79 |
| Filipino | Malay | 61.72 | 57.88 | 55.41 | 58.17 | 61.58 | 21.31 | 51.29 | 50.00 | 46.32 | 13.44 | 48.53 | 53.37 | 38.92 | 49.63 | 63.77 | 63.76 |
| Filipino | Khmer | 37.65 | 3.02 | 34.33 | 39.91 | 43.48 | 13.11 | 26.19 | 14.83 | | 23.84 | 17.57 | 30.43 | 2.35 | 11.78 | 43.59 | 42.98 |
| Filipino | Lao | 44.13 | 4.30 | 40.05 | 3.36 | 49.64 | 10.73 | 26.07 | 14.26 | 11.04 | 2.88 | 10.25 | 25.94 | 3.51 | 10.24 | 49.72 | 51.63 |
| Filipino | Burmese | 41.00 | 0.72 | 42.80 | 44.86 | 47.67 | 17.72 | 29.42 | 32.35 | 9.97 | 35.23 | 16.86 | 25.91 | 16.33 | 12.08 | 44.17 | 45.98 |
| Filipino | Indonesian | 64.77 | 58.88 | 57.86 | 58.64 | 62.25 | 21.08 | 53.45 | 53.66 | 53.46 | 13.68 | 51.19 | 55.47 | 41.96 | 50.88 | 62.33 | 63.02 |
| Filipino | Vietnamese | 55.96 | 52.48 | 54.48 | 51.84 | 53.71 | 14.38 | 47.83 | 49.48 | 46.47 | 8.51 | 45.87 | 48.03 | 30.29 | 40.98 | 69.46 | 67.49 |
| Filipino | English | 69.83 | 65.23 | 67.07 | 63.32 | 68.54 | 19.50 | 63.28 | 64.96 | 56.53 | 19.86 | 62.92 | 63.68 | 53.91 | 62.69 | 68.49 | 67.70 |
| Filipino | Chinese | 33.77 | 24.61 | 32.59 | 32.88 | 32.44 | 3.11 | 29.12 | 29.54 | 23.52 | 5.27 | 25.40 | 25.91 | 17.26 | 28.74 | 38.52 | 30.26 |
| Indonesian | Thai | 54.04 | 53.80 | 53.81 | 52.75 | 51.67 | 13.47 | 48.42 | 49.46 | 27.09 | 15.13 | 44.18 | 45.05 | 23.74 | 41.73 | 54.29 | 50.39 |
| Indonesian | Malay | 63.05 | 64.02 | 61.33 | 61.08 | 59.99 | 21.63 | 57.28 | 55.40 | 51.87 | 26.08 | 53.86 | 54.05 | 45.13 | 54.42 | 59.28 | 59.81 |
| Indonesian | Khmer | 38.03 | 3.06 | 34.98 | 40.62 | 43.86 | 12.51 | 28.70 | 16.49 | 11.96 | 33.56 | 18.54 | 31.44 | 2.38 | 12.10 | 43.71 | 43.86 |
| Indonesian | Lao | 44.92 | 4.37 | 40.34 | 3.34 | 49.88 | 10.84 | 28.57 | 18.12 | 11.40 | 3.56 | 10.70 | 25.95 | 4.91 | 10.58 | 48.43 | 49.68 |
| Indonesian | Burmese | 39.74 | 0.74 | 43.01 | 44.92 | 47.61 | 17.74 | 30.47 | 36.80 | 9.74 | 35.53 | 17.05 | 26.71 | 15.57 | 11.73 | 43.90 | 46.35 |
| Indonesian | Filipino | 55.76 | 18.46 | 53.84 | 54.60 | 56.55 | 22.48 | 46.61 | 51.09 | 40.23 | 22.55 | 48.60 | 50.58 | 37.14 | 42.64 | 54.69 | 55.27 |
| Indonesian | Vietnamese | 56.09 | 57.30 | 55.88 | 53.69 | 53.99 | 14.13 | 51.83 | 53.25 | 54.24 | 17.63 | 50.68 | 49.94 | 32.68 | 47.29 | 58.64 | 57.40 |
| Indonesian | English | 68.76 | 69.61 | 68.91 | 64.53 | 68.30 | 19.29 | 66.24 | 67.76 | 66.56 | 23.37 | 58.49 | 64.47 | 58.53 | 67.42 | 69.08 | 68.05 |
| Indonesian | Chinese | 34.31 | 30.61 | 35.10 | 34.69 | 33.35 | 3.17 | 32.39 | 32.25 | 31.22 | 8.05 | 28.97 | 27.71 | 19.90 | 33.13 | 36.69 | 32.47 |
| Vietnamese | Thai | 52.08 | 51.11 | 51.82 | 50.84 | 49.69 | 13.41 | 46.88 | 47.69 | 26.84 | 7.92 | 42.06 | 44.06 | 23.52 | 40.36 | 55.90 | 53.40 |
| Vietnamese | Malay | 58.92 | 59.02 | 57.46 | 57.08 | 58.73 | 21.97 | 53.08 | 54.03 | 49.44 | 12.70 | 48.65 | 52.94 | 39.31 | 50.69 | 58.59 | 54.58 |
| Vietnamese | Khmer | 36.87 | 2.98 | 34.46 | 39.68 | 42.72 | 12.91 | 28.42 | 16.05 | 11.93 | 32.52 | 18.62 | 29.75 | 2.93 | 11.56 | 42.94 | 42.02 |
| Vietnamese | Lao | 43.71 | 4.31 | 39.96 | 3.28 | 48.54 | 11.03 | 27.71 | 17.35 | 11.17 | 3.78 | 11.07 | 27.03 | 6.03 | 10.78 | 48.12 | 48.98 |
| Vietnamese | Burmese | 39.81 | 0.72 | 43.29 | 44.72 | 46.84 | 17.83 | 30.05 | 36.46 | 9.88 | 33.16 | 16.80 | 25.80 | 16.36 | 11.48 | 43.78 | 45.22 |
| Vietnamese | Filipino | 55.26 | 18.16 | 51.78 | 52.85 | 54.25 | 22.62 | 45.02 | 50.23 | 39.67 | 19.97 | 47.24 | 48.34 | 35.28 | 41.45 | 51.89 | 52.75 |
| Vietnamese | Indonesian | 61.64 | 60.49 | 60.78 | 58.06 | 59.32 | 21.56 | 57.20 | 59.28 | 59.57 | 13.06 | 52.69 | 55.07 | 43.01 | 54.59 | 61.96 | 59.98 |
| Vietnamese | English | 63.04 | 64.11 | 62.96 | 59.94 | 62.62 | 19.55 | 61.34 | 61.74 | 61.13 | 18.28 | 47.38 | 58.74 | 51.99 | 61.49 | 66.42 | 64.50 |
| Vietnamese | Chinese | 32.58 | 30.33 | 33.39 | 33.04 | 31.93 | 3.21 | 30.84 | 30.32 | 30.43 | 7.67 | 26.61 | 26.31 | 19.65 | 31.17 | 34.86 | 29.54 |
| English | Thai | 57.35 | 58.17 | 57.66 | 55.98 | 55.67 | 13.53 | 51.56 | 54.39 | 28.15 | 26.87 | 47.61 | 47.63 | 24.60 | 44.77 | 61.71 | 56.87 |

| Source | Target | Model chrF Scores | | | | | | | | | | | | | | | |
|--------|--------|------|-------|--------|------|------|-------|------|-------|------|------|--------|-------|-------|-------|--------|------|
| | | Gem3 | SeedX | QwenMT | Huny | MtPO | Qw2.5 | Qw3 | Apert | Aya | Emma | LLa3.1 | LLaX3 | Mistr | Tower | Google | NLLB |
| English | Malay | 67.96 | 70.11 | 66.79 | 64.40 | 68.27 | 21.84 | 61.20 | 64.03 | 55.77 | 38.52 | 61.91 | 61.31 | 45.59 | 57.47 | 65.27 | 64.87 |
| English | Khmer | 39.25 | 3.17 | 36.00 | 42.44 | 47.24 | 13.15 | 30.25 | 17.79 | 11.99 | 42.76 | 19.01 | 32.37 | 2.25 | 12.65 | 47.22 | 47.37 |
| English | Lao | 47.45 | 4.52 | 43.97 | 3.48 | 54.81 | 11.35 | 30.98 | 19.74 | 11.37 | 5.37 | 9.94 | 26.79 | 4.64 | 10.88 | 53.14 | 55.14 |
| English | Burmese | 41.62 | 0.77 | 46.01 | 47.04 | 50.92 | 18.73 | 32.46 | 39.95 | 9.93 | 37.92 | 17.86 | 28.75 | 12.89 | 12.35 | 47.75 | 49.78 |
| English | Filipino | 62.69 | 18.89 | 58.03 | 58.48 | 61.49 | 22.15 | 50.48 | 57.25 | 42.64 | 24.45 | 55.21 | 54.44 | 40.19 | 46.58 | 61.76 | 63.47 |
| English | Indonesian | 72.11 | 71.95 | 71.63 | 65.97 | 69.35 | 21.48 | 66.43 | 70.45 | 69.16 | 27.21 | 66.70 | 62.73 | 49.80 | 64.17 | 72.87 | 71.32 |
| English | Vietnamese | 60.22 | 63.73 | 60.76 | 57.44 | 59.39 | 14.77 | 56.80 | 59.15 | 58.75 | 25.15 | 56.24 | 53.56 | 35.58 | 50.93 | 67.26 | 65.16 |
| English | Chinese | 38.76 | 35.69 | 40.14 | 40.23 | 38.92 | 3.27 | 37.68 | 36.97 | 36.09 | 16.85 | 33.68 | 31.56 | 26.24 | 39.93 | 41.98 | 35.57 |
| Chinese | Thai | 50.65 | 50.32 | 50.31 | 50.21 | 49.10 | 13.79 | 46.00 | 47.08 | 25.68 | 11.30 | 40.72 | 42.39 | 23.35 | 39.82 | 53.57 | 53.82 |
| Chinese | Malay | 56.00 | 56.44 | 54.78 | 55.34 | 55.66 | 21.94 | 50.57 | 52.21 | 46.90 | 15.25 | 49.21 | 49.16 | 37.00 | 48.52 | 56.57 | 57.11 |
| Chinese | Khmer | 35.26 | 2.79 | 34.32 | 38.86 | 41.11 | 13.39 | 27.16 | 15.29 | 11.82 | 36.97 | 17.32 | 27.37 | 1.85 | 12.06 | 42.07 | 42.62 |
| Chinese | Lao | 40.85 | 3.98 | 37.65 | 2.75 | 45.54 | 11.64 | 26.61 | 16.13 | 10.79 | 2.59 | 10.47 | 22.86 | 3.22 | 10.74 | 46.14 | 47.79 |
| Chinese | Burmese | 37.95 | 0.65 | 43.24 | 44.16 | 46.27 | 18.41 | 31.30 | 37.31 | 9.82 | 32.06 | 16.70 | 25.56 | 16.92 | 12.17 | 43.61 | 46.88 |
| Chinese | Filipino | 52.46 | 17.92 | 49.42 | 51.45 | 51.48 | 22.91 | 43.23 | 48.04 | 38.08 | 19.01 | 45.73 | 45.35 | 34.67 | 40.30 | 53.40 | 54.19 |
| Chinese | Indonesian | 58.66 | 58.23 | 58.42 | 56.95 | 56.85 | 21.71 | 54.98 | 56.36 | 56.91 | 16.83 | 52.87 | 51.12 | 40.11 | 52.95 | 58.87 | 58.69 |
| Chinese | Vietnamese | 51.93 | 53.63 | 51.93 | 51.14 | 50.58 | 14.12 | 49.15 | 49.83 | 50.40 | 10.73 | 46.70 | 45.82 | 30.03 | 44.79 | 54.92 | 53.85 |
| Chinese | English | 59.24 | 59.47 | 59.95 | 58.50 | 59.34 | 19.42 | 58.82 | 58.41 | 58.19 | 25.99 | 56.75 | 54.82 | 51.87 | 60.06 | 60.59 | 61.49 |

## A.9 CASE STUDY: TRANSLATION BEHAVIOR ANALYSIS

To illustrate the diversity of translation behaviors observed during data curation, Table 8 presents a concrete example where the same Vietnamese idiom is translated by four different assistant configurations with varying decoding strategies.

| Role | Utterance |
|------|-----------|
| System | You are a helpful assistant. |
| User | Please translate the following sentence into English: *Cho ngap phai ruoi* |
| AssistantA | By sheer chance. |
| AssistantB | A dog yawns and catches a fly. |
| AssistantC | The dog yawned and hit a fly (Literal meaning: when the dog yawns, it bumps into a fly.) This is a metaphor indicating that while doing something you intended to avoid or miss someone or something, you unexpectedly ran into him or it. In other words, you tried to steer clear but ended up encountering it anyway. Comparable English idioms include "things go contrary to one's wishes" or "the more you try to hide it, the more obvious it becomes," both signalling intention-outcome mismatch. |
| AssistantD | The dog hit a fly, the dog hit a fly—yes, it keeps hitting the same fly. That is literally what it means, it just keeps saying the dog is hitting the fly again and again. (Repetitive and partially incorrect.) |

**Table 8:** Example conversation showing four different translation approaches: idiomatic, literal, explanatory, and noisy outputs in response to the same Vietnamese idiom.

### A.9.1 TRANSLATION QUALITY PATTERNS

Table 8 illustrates four distinct translation approaches, revealing key patterns in model behavior:

**Optimal Translation (AssistantA):** The response "By sheer chance" captures the idiomatic meaning effectively while maintaining brevity. This represents the ideal translation—accurate and succinct without unnecessary elaboration.

**Literal Translation (AssistantB):** "A dog yawns and catches a fly" preserves the original structure but fails to convey the cultural meaning of the idiom, potentially confusing readers unfamiliar with Vietnamese expressions.

**Over-explanation (AssistantC):** This response exemplifies the overgeneration problem frequently associated with high entropy outputs. Despite being informative, the 72-word response significantly

diverges from the core translation objective. The model's uncertainty manifests as excessive elaboration.

**Degraded Output (AssistantD):** Features inaccuracies, repetition, and self-dialogue patterns. This output typifies problems associated with excessive response length and high entropy, where the model becomes trapped in loops of self-correction and redundant explanation.

### A.9.2 ENTROPY-LENGTH CORRELATION

Our analysis reveals a strong correlation between response length and translation quality. Optimal translations efficiently convey meaning without superfluous context or explanation. Longer responses often coincide with increased token entropy, indicating model uncertainty that manifests as verbose outputs.

These patterns align with our broader findings that entropy loss and response length serve as key diagnostic indicators for translation quality. When models generate unnecessarily long responses, they typically exhibit higher entropy across generated tokens—a sign of uncertainty that degrades translation performance. MtPO addresses this by monitoring and constraining these metrics, guiding models toward concise, accurate translations like AssistantA rather than verbose or literal approaches.

### A.10 KL DIVERGENCE APPROXIMATION ANALYSIS

In our entropy diagnostics, we employ three Monte-Carlo estimators—k1, k2, and k3—to approximate KL divergence between policy and reference distributions. Each estimator offers a distinct bias-variance profile when computed from sampled translations.

### A.10.1 METHOD COMPARISON

**k1 Method (Naive Estimator):** Directly uses the negative expectation of log ratios:

$$k1 = -\mathbb{E}_{x \sim q} \left[ \log \frac{p(x)}{q(x)} \right] \tag{15}$$

This estimator is unbiased but exhibits high variance and may yield negative values on small batches even though KL divergence is non-negative.

**k2 Method (Squared Log Ratio):** Approximates KL divergence using the squared log ratio:

$$k2 = \frac{1}{2}\mathbb{E}_{x \sim q} \left[ \left( \log \frac{p(x)}{q(x)} \right)^2 \right] \tag{16}$$

It introduces bias yet typically maintains lower variance. k2 corresponds to an f-divergence whose second-order expansion matches KL divergence when $p$ is close to $q$.

**k3 Method (Bregman Divergence):** Evaluates the expectation of an exponential transform:

$$k3 = \mathbb{E}_{x \sim q} \left[ \frac{p(x)}{q(x)} - 1 - \log \frac{p(x)}{q(x)} \right] \tag{17}$$

This estimator remains unbiased with comparatively low variance, measuring the vertical gap between $\log(x)$ and its tangent approximation.

### A.10.2 PRACTICAL SELECTION

Choosing among these estimators depends on the desired bias-variance trade-off:

- **k1**: Unbiased but high variance; sensitive to entropy spikes.
- **k2**: Slightly biased yet low variance; reliable near convergence.
- **k3**: Unbiased with low variance; preferred default in our monitoring stack.

For MtPO's training diagnostics, k2 and k3 deliver smoother curves than k1, particularly under entropy collapse scenarios that arise during reinforcement learning. These approximation methods build on established Monte Carlo techniques(Joschu, 2020). Their stability provides more trustworthy signals when tuning entropy regularization and reward scaling.

## A.11 Language Model Assistance in Paper Preparation

In the preparation of this manuscript, we utilized Large Language Models (LLMs) to assist with language polishing and writing optimization, following emerging practices in academic writing assistance. Specifically, LLMs were employed for:

- **Language fluency improvement**: Enhancing sentence structure and expression to improve readability and adherence to academic writing standards
- **Grammar and spelling verification**: Identifying and correcting potential grammatical errors and spelling issues
- **Terminology consistency**: Ensuring consistent usage of technical terms throughout the manuscript
- **Clarity enhancement**: Improving the articulation of complex concepts to make them more accessible and comprehensible

This section contains additional information and supplementary materials. It is important to emphasize that all core technical content, experimental design, data analysis, and scientific conclusions represent original work by the authors. LLMs were used solely as language polishing tools and did not participate in any substantive research content creation or formation of academic viewpoints. All technical contributions and innovations in this research stem entirely from the independent research work of the author team.

We believe that the reasonable use of advanced language technology tools to enhance academic writing quality, while maintaining academic integrity, represents beneficial practice that facilitates better communication of research findings and promotes scholarly exchange.

