# OpenReview forum: "Extending Foundation Models to Low-Resource Languages: Vocabulary Expansion and Policy Optimization"
_ICLR.cc/2026/Conference — ICLR 2026 Conference Desk Rejected Submission_

### Official Review · Reviewer_218C · 2025-10-27

**Soundness:** 2
**Presentation:** 2
**Contribution:** 2
**Rating:** 4
**Confidence:** 3

**Summary:**

The work introduces multilingual translation policy optimization, which i) expands the vocabulary of tokenizer, ii) balances mixing ratio in training corpora and iii) reinforcement learning with verifiable rewards. i) directly improves inference efficiency for low-resource languages. ii) preserves model capability in English and iii) controls length, format, target language and language consistency in the output. The objective also includes an token-position based entropy regularizer to encourage more exploration in the early stages of decoding. Overall, the approach demonstrates token efficiency, reduces token entropy, improves length stability and token diversity. Translation performance also improves in terms of sacreBLEU.

**Strengths:**

1. The paper describes a comprehensive pipeline to adapt LLM for low-resource translation.
2. The expanded VLRL with entropy regularized objective is interesting, and potentially inspires many questions to explore.
3. The approach demonstrates clear advantage in terms of entropy increment, token diversity and length stability.

**Weaknesses:**

1. Line 187: "Production translation systems encounter frequent failure modes: overlong outputs, format corruption, language mixing, and off-target responses... " is there a citation for this?
2. While the results in Table 2 shows marginal improvement of MtPO over SFT baseline, there is no evidence that any of this is a direct result of more length stability, format and language accuracy. It could be possible that vocabulary expansion alone improves translation, or that superior format accuracy (Table 1) alone makes translation better.
3. Increasing token entropy leads to less confident models, what is the evidence that this is beneficial specifically for translations? And why not start with highly confident tokens, then explore only towards the end of the sequence, as in https://arxiv.org/pdf/2507.01679?
4. Again, the effect of length stability on improved translation performance is unclear. The right length ratio often depends on the source and target languages. The values would be vastly different, for instance, between En-Bn and Bn-En, given their vocab difference. How is this being accounted for in the current set up?
5. Please also report translation performance in terms of chrF, which better captures morphological variability, especially in lower-resourced languages.
6. The derivations in Section 5 don't seem to help in our understanding of why any of the properties make sense to improve translation.
7. The overall presentation is poor. There are no labels and captions for figures.

**Questions:**

1. Table 1 shows MtPO stands out in terms of format accuracy, relative to the other models. What if format accuracy alone account for all improvement of MtPO in Table 2? What if more meaningful tokens given vocabulary expansion alone drove most of the improvement instead? Please consider adding ablation studies.


See also points 3 and 4 above.

---

> ### Author Response · Authors · 2025-11-17
> **Rebuttal1117_3**
>
> **General Response**
> First of all, thank you for the thorough review and the valuable feedback—we have adjusted our paper’s layout and clarified several technical points in response to your comments.
>
> **Q1: Line 187: Citation for "Production translation systems encounter frequent failure modes..."?**
>
> **Response:** Yes—we’ve seen all four failure modes in production deployments, and each has been documented in prior work:
> *   **Mixed/Contaminated Citations:** *Arora et al. (2024)* (arXiv:2406.20052)
> *   **Off-target Translations:** *Zhang et al. (2023)* (Findings of ACL 2023)
> *   **Repetition Loops:** *Xu et al. (2024)* (EMNLP 2024)
> *   **Format Corruption:** *Freitag et al. (2020)* (WMT 2020)
>
> **Q2: Is the improvement in Table 2 due to length stability/format accuracy, or just vocabulary/SFT?**
>
> **Response:** First, MtPO is a full three-stage pipeline—not just an alignment tweak—so the gains you see in Table 2 reflect the combined effect of CPT, SFT, and alignment rather than a single component.
> *   **Role of Vocabulary/SFT:** Vocabulary expansion and SFT already reduce token-length disparities and preserve general abilities.
> *   **Role of Alignment (RLVR):** The alignment phase adds two safeguards. RLVR’s four verifiers (format, language, length, mixing) do not directly “boost translation accuracy,” but they filter low-quality rollouts so the policy receives clean feedback—without them, PPO/GRPO-style RL quickly produces overlong, off-target, or malformed outputs, which would tank BLEU.
> *   **Role of Entropy:** Dynamic entropy control then encourages better paraphrasing only when the verifier filters pass, making sure the exploration stays within valid translations.
>
> These four diagnostics are exactly the issues we see in production, so keeping them accurate is essential to convert the vocabulary/SFT improvements into a stable end model.
>
> **Q3: Why increase token entropy? Why not start confident and explore later (like arXiv:2507.01679)?**
>
> **Response:** Entropy here isn’t about ending with a “less confident” translator; it’s a training-time device to avoid the well-known failure where DAPO/GRPO collapse to one high-reward pattern and then emit repetitive, overlong text (see Appendix A9.1 demos).
> *   **Our Strategy:** We drive entropy high only at the start of the sequence, then decay it token by token. This allows the model to explore lexical/structural variants while the context is still being set, and gradually sharpens as it finishes the sentence. This aligns with translation practice—word order and discourse markers benefit from early exploration, but later tokens should finalize the phrasing.
> *   **Comparison:** The paper you cite (end-of-sequence exploration) tackles the opposite regime: it begins confidently and perturbs the tail, which helps creative tasks but in our experiments caused “locked-in” openings that never recover if they go off-target. We can add a short paragraph comparing the two strategies and cite that work explicitly so this trade-off is clear.

---

> ### Author Response · Authors · 2025-11-17
> **Rebuttal1117_3.1**
>
> **Q4: How is length stability accounted for given varying token ratios (e.g., En-Bn vs Bn-En)?**
>
> **Response:** Length control is a critical point, and we illustrate it with examples in the appendix.
> *   **Tokenizer Optimization:** After optimizing the tokenizer, we no longer see low-compression mappings from one language to another. We gathered roughly 4,000 common words for each of the ten ASEAN languages, so after CPT+SFT the model already avoids large token-ratio disparities between source and target texts. Section 4.1 of the Experiments highlights exactly how the optimized vocabulary delivers those gains.
> *   **Flexible Constraints:** RLVR doesn’t hard-clamp the ratio either—we keep a flexible 0.2–2.0 window, and any sample falling within that band is retained for training.
>
> **Q5: Please report translation performance in terms of chrF.**
>
> **Response:** Thank you for the feedback. We have added the **chrF** benchmark as another translation-quality metric and, on top of that, included **COMET** as well as non-LLM baselines such as **Google Translate** and **NLLB**. All results have been updated in **Table 3**, with the full chrF breakdown moved to the appendix.
>
> **Q6: The derivations in Section 5 don't seem to help understanding.**
>
> **Response:** The analysis section is meant to clarify the design of the alignment algorithm. We have restructured the paper:
> *   **Consolidation:** We moved the pseudocode out of the appendix and into the Experiments section, and we folded the analysis content into Methods so everything sits next to the alignment description.
> *   **RLVR Mechanism:** In RLVR, the goal is to filter rollouts that meet hard constraints so the RL stage only trains on high-quality sequences. During alignment we apply several adjustments:
>     1.  **Explicit entropy-boosting schedule:** Decays with sequence length—early tokens explore more and favor paraphrasing, but once a strategy is chosen it stays precise.
>     2.  **Temperature consistency:** Between training and inference so rollout behavior matches deployment.
>     3.  **Batch-level advantage normalization:** Switched from group-level to reduce variance.
> *   **Validation:** We also run KL ablations to confirm stability. The emphasis on format accuracy comes from production and ablation findings: algorithms like GRPO easily trigger extreme length growth or code-mixing, and Appendix demos show those failure cases.
>
> **Q7: What if format accuracy or vocabulary expansion alone accounts for all improvement? Please consider adding ablation studies.**
>
> **Response:**
> *   **Full Pipeline:** MtPO is a three-stage training pipeline. The gains reflect the combined effect of pretraining (CPT), supervised fine-tuning (SFT), and alignment (MtPO).
> *   **Role of Format:** Ensuring correct formatting helps filter low-quality rollouts so the alignment stage receives cleaner signals, but formatting itself is not the sole source of translation gains.
> *   **Ablation Study:** To explicitly quantify these contributions, we added an ablation study to **Table 3**:
>
> | Setting | Score | $\Delta$ | Role |
> | :--- | :--- | :--- | :--- |
> | w/o CPT | 33.3 | **-2.8** | Foundation |
> | SFT Only | 35.4 | **-0.7** | Alignment |
> | **Full** | **36.1** | - | Refinement |
>
> **Analysis:**
> 1.  **Necessity of CPT:** The significant drop (**-2.8**) confirms vocabulary expansion/CPT is foundational for linguistic adaptation.
> 2.  **Gain from MtPO:** The full model consistently outperforms the SFT baseline (e.g., +1.3~1.8 on specific tasks), proving that MtPO provides essential alignment optimization beyond just vocabulary or format fixes.

---

> > ### Author Response · Authors · 2025-11-17
> > **We are retraining an SFT/alignment model from Qwen-base as supporting evidence for your ablation study.**
> >
> > We are retraining an SFT/alignment model from Qwen-base as supporting evidence for your ablation study.

---

> ### Author Response · Authors · 2025-11-21
> **Translation Performance Comparison (chrF Scores)**
>
> **Table: Translation Performance Comparison (chrF Scores)**
>
> | Model | xx→en | en→xx | xx→xx | Avg. |
> |-------|-------|-------|-------|------|
> | **Multilingual Chat Models** | | | | |
> | Gemma3-27B-IT | 62.87 | 54.15 | 47.53 | 49.73 |
> | Qwen3-8B | 58.45 | 46.42 | 36.80 | 39.93 |
> | Qwen2.5-7B-Instruct | 19.41 | 15.58 | 14.50 | 15.10 |
> | Apertus-8B-Instruct | 60.51 | 46.63 | 36.86 | 40.22 |
> | Tower-Plus-9B | 54.73 | 37.74 | 29.33 | 32.74 |
> | **Translation-Focused Models** | | | | |
> | Qwen-MT-Plus | 60.10 | 53.44 | 45.79 | 48.19 |
> | Seed-X-PPO-7B | 51.61 | 36.33 | 24.72 | 28.57 |
> | Hunyuan-MT-7B | 55.24 | 48.38 | 39.53 | 41.99 |
> | **Translation-Focused without LLM** | | | | |
> | Google Translate | **65.68** | 55.70 | 47.94 | 50.49 |
> | NLLB-200 | 62.95 | 52.13 | 44.92 | 47.45 |
> | **Our Models** | | | | |
> | Light-TLLM-7B-MtPO w/o CPT | 56.82 | 51.82 | 43.60 | 46.86 |
> | Light-TLLM-7B-SFT | 59.82 | 52.94 | 44.46 | 48.26 |
> | Light-TLLM-7B-MtPO | 62.74 | **56.22** | **48.80** | **50.94** |
>
> Moreover, we performed post-training on a model variant without CPT (Continual Pre-Training) and incorporated the comparative results in the table at the appropriate location.

---

> ### Author Response · Authors · 2025-11-26
> **Why do models trained with GRPO tend to generate longer responses? How is this bias addressed?**
>
> Here is the translation of your text into English, suitable for a rebuttal or academic context:
>
> ***
>
> # **Why do models trained with GRPO tend to generate longer responses? How is this bias addressed?**
>
> First, let us examine the GRPO objective function and its normalization formula:
>
> $$
> J_{GRPO} = E \left[ \frac{1}{G} \sum_{i=1}^G \textcolor{red}{\frac{1}{|o_i|}} \sum_{t=1}^{|o_i|} \min( r_t A_{i,t}, \text{clip}(r_t, \dots) A_{i,t} ) \right]
> $$
>
> $$
> \hat{A}_{i,t} = \frac{R(\mathbf{q}, \mathbf{o}_i) - \text{mean}(\{R(\mathbf{q}, \mathbf{o}_1),\ldots,R(\mathbf{q}, \mathbf{o}_G)\})}{\color{red}{\text{std}(\{R(\mathbf{q}, \mathbf{o}_1),\ldots,R(\mathbf{q}, \mathbf{o}_G)\})}},
> $$
>
> The parts highlighted in red represent two biases inherent in GRPO. Here, bias refers to the differential treatment of variance among samples of varying lengths:
>
> *   **Response-level Length Bias:** For positive advantages, this bias causes **shorter responses to receive larger gradient updates**, thereby encouraging the policy to **prioritize more concise expressions** for correct answers. Conversely, for negative advantages, since longer responses have a larger $|\mathbf{o}_i|$, they receive smaller penalties. This leads the policy to favor **longer responses** when generating incorrect answers.
>
> *   **Question Difficulty-level Bias:** Questions with lower standard deviations (e.g., questions that are too simple or too difficult, where rewards are almost entirely 1 or 0) are assigned higher weights during policy updates. This **question-level normalization** assigns varying weights to different questions within the objective function, thereby introducing a **difficulty bias** during the optimization process.
>
> While this approach might initially appear reasonable—since the loss for both long and short samples is optimized as a mean at the token level, suggesting no inherent length bias—closer inspection reveals an issue.
>
> Due to normalization, the gradient contribution of each sample is equalized within the batch. However, as noted in the GRPO paper, the advantage function is calculated for the entire sample, not at the token level.
>
> This implies that long samples and short samples share the same advantage magnitude. Consequently, tokens in shorter samples receive stronger gradient updates because there are fewer tokens to "share" the gradient. This elucidates why the model struggles to effectively suppress long, incorrect answers.
>
> Regarding the first bias, GRPO averages the loss within the sequence first and then averages across samples. This results in **important tokens in long sequences being diluted**, while simultaneously failing to impose sufficient penalties on verbose, low-quality outputs.
> low-quality verbose outputs receive significant penalties, thereby **maintaining the normal order of sequence length growth**.
>
> Ultimately, this design leads to the phenomenon where GRPO induces the model to produce longer outputs. This is particularly pronounced in translation tasks, where the model often follows a complete translation with extensive noun explanations or synonymous phrasings.

---

> ### Author Response · Authors · 2025-11-27
> **About  https://arxiv.org/pdf/2507.01679  ICLR20206 Submission Number: 4785**
>
> Thank you for the reference. Since this paper appeared on arXiv in July 2025 and is also an ICLR 2026 submission (ID: 4785), we consider it concurrent work. We agree it is an interesting approach and will add a citation and a comparative discussion in our revised manuscript to clarify the differences between our works.

---

### Official Review · Reviewer_FtdW · 2025-10-31

**Soundness:** 4
**Presentation:** 2
**Contribution:** 3
**Rating:** 6
**Confidence:** 5

**Summary:**

The presents method to extend large language models for low resource translation via (1) vocabulary expansion, (2) fine-tuning with distilled data, and (3) a novel reinforcement learning method. This mainly follows the current state-of-the-art, it is solid work, and the particular reinforcement learning method is novel.

**Strengths:**

Solid experimentation with 8 language pairs, many of the low resource. This is generally well executed work with covincing results.

Novel variant of reinforcement learning (MtPO) that introduces a number of new elements. The method needs to be explained better.

The vocabulary expansion is executed well, but it is not a novel method. You may want to emphasize this less.

**Weaknesses:**

The write-up of the paper needs to be improved substantially. Key elements are not explained or explained insuffienctly.

Section 3.3.2: This section needs to be restructured, so that you first motivate the approach, introduce the terms, and then have the complete equations. It is currently difficult to understand. Since this is maybe the strongest contribution, you need to explain it.

The "analysis" method is more a motivation for the approach - it should be in the methods section.

Minor issues:
line 68: "falls short in production settings, where reliance on fixed translation-style instructions results in overly narrow and inflexible use cases" -> can you explain this more and add a citation?

Line 191ff: connect the terms in the equation to the text

Line 310: This is the first time you mention "KL-control schemes (K2, K3)" - I assume that this is KL-divergence regularization, but you need to explain it.

Figure 3ff: None of these figures are labeled or have a caption.

Line 326: Is "entropy collapse" overfitting to the training data?

**Questions:**

I do not understand the long discussion of length inflation. In machine translation, the length is fixed within some margin, so this should not happen.

---

> ### Author Response · Authors · 2025-11-17
> **Rebuttal1117_2**
>
> **General Response**
> We sincerely thank the reviewer for the thorough review and valuable feedback. We have significantly improved the paper's structure and clarity, particularly by overhauling Section 3 and rewriting Section 4 to better explain our methodology and experimental setup.
>
> ---
>
> **Q1: Improve the write-up substantially; key elements are insufficiently explained.**
>
> **Response:** We have conducted a major revision to enhance clarity and cohesion:
> 1. **Pipeline Restructuring:** Section 3 is now reorganized into three distinct subsections: *Tokenizer Expansion*, *Continued Pre-training*, and *RLVR-based Policy Optimization*.
> 2. **Visual Aids:** We added a workflow diagram and pseudocode to clearly illustrate data flow, hyperparameters, and update rules.
> 3. **Detailed Definitions:** The RLVR subsection now explicitly defines each verifier, thresholds, and reward contributions. Full formulas and concrete examples are provided in Appendix A.
> 4. **Experimental Clarity:** Section 4 has been rewritten to specify datasets, splits, and explicitly link each figure/table to the research questions (efficiency, training effects, end-task gains).
>
> **Q2: Restructure Section 3.3.2 (motivate, introduce terms, then equations).**
>
> **Response:** We agree that this section is central to our contribution. We have restructured it as follows:
> 1. **Motivation:** We first explain the necessity of the three-stage flow: CPT for vocabulary efficiency, SFT with curriculum learning for general capabilities, and MtPO for alignment.
> 2. **Formalization:** We moved key formulas from the appendix to the main text to ensure the methodology is self-contained.
> 3. **Technical Details:** We explicitly explain the stability mechanisms, including rollout temperature consistency, position-decaying entropy, and batch-level advantage normalization.
>
> **Q3: Explain "falls short in production settings" (line 68) and add citations.**
>
> **Response:** Existing translation-focused models often rely on rigid, fixed templates. For example, models like **Qwen-MT-Plus** or **Seed-X** require specific prompts encoding source/target languages, which limits their flexibility in mixed-language scenarios or streaming applications.
> *   **Our Advantage:** MtPO automatically detects eight low-resource languages while accepting arbitrary user prompts, offering superior flexibility.
> *   **Citations:** We have added citations to the *Hunyuan-MT Technical Report* (Zheng et al., 2025) and *Seed-X* (Cheng et al., 2025) to support this discussion.

---

> ### Author Response · Authors · 2025-11-17
> **Rebuttal_2.1**
>
> **Q1: Line 310: Explain "KL-control schemes (K2, K3)".**
>
> **Response:** Correct—those experiments are precisely KL-regularization ablations. KL is a key RL hyperparameter with multiple standard controllers. Because MtPO adds explicit entropy boosts during alignment, the policy can collapse unless the KL budget is chosen carefully. Following PPO-K2 and GRPO-K3, we evaluate three regimes:
> *   **K2:** Moving-target KL budget.
> *   **K3:** Strict fixed budget.
> *   **No-KL:** Relies solely on entropy regularization.
>
> These comparisons demonstrate that MtPO’s entropy-preserving design maintains stability even under high exploration, justifying our final choice of the **No-KL** variant (relying solely on entropy regularization).
>
> **Q2: Line 326: Is "entropy collapse" overfitting to the training data?**
>
> **Response:** "Entropy collapse" refers to **distributional degeneration** during policy-gradient training: mis-specified KL constraints or advantage scaling cause token probabilities to concentrate on a few patterns, sharply reducing exploration.
> *   **Distinction:** This is different from memorizing the training set—it can happen even on unseen prompts because it’s a **stability issue**, not an experimental flaw.
> *   **Diagnosis:** We track entropy curves and KL diagnostics to detect it: once entropy drops, the model produces overlong, repetitive translations on new inputs. Appendix KL analyses illustrate why we add position-dependent entropy regularization and carefully tuned KL budgets, rather than indicating any data leakage.
>
> **Q3: Discussion of length inflation in translation.**
>
> **Response:** Appendix A9.1 (“Length Diagnostics Demo”) documents concrete cases: with the same English sentence, GRPO under a loose KL budget generates outputs **3–4× longer**—translating first, then appending long noun explanations or even continuing the narrative; in worst cases it repeats segments. These failures motivated two MtPO design choices:
> 1.  **Position-decaying entropy bonus:** Early tokens stay diverse, but entropy automatically shrinks later, preventing divergence.
> 2.  **Length-aware constraints:** We introduce length-aware normalization plus an RLVR length verifier to stop RL (especially GRPO variants) from inflating outputs via reward coupling.
>
> Appendix A9.1 screenshots/logs make these behaviors and fixes explicit.

---

> ### Author Response · Authors · 2025-11-26
> **Why do models trained with GRPO tend to generate longer responses? How is this bias addressed?**
>
> Here is the translation of your text into English, suitable for a rebuttal or academic context:
>
> ***
>
> # **Why do models trained with GRPO tend to generate longer responses? How is this bias addressed?**
>
> First, let us examine the GRPO objective function and its normalization formula:
>
> $$
> J_{GRPO} = E \left[ \frac{1}{G} \sum_{i=1}^G \textcolor{red}{\frac{1}{|o_i|}} \sum_{t=1}^{|o_i|} \min( r_t A_{i,t}, \text{clip}(r_t, \dots) A_{i,t} ) \right]
> $$
>
> $$
> \hat{A}_{i,t} = \frac{R(\mathbf{q}, \mathbf{o}_i) - \text{mean}(\{R(\mathbf{q}, \mathbf{o}_1),\ldots,R(\mathbf{q}, \mathbf{o}_G)\})}{\color{red}{\text{std}(\{R(\mathbf{q}, \mathbf{o}_1),\ldots,R(\mathbf{q}, \mathbf{o}_G)\})}},
> $$
>
> The parts highlighted in red represent two biases inherent in GRPO. Here, bias refers to the differential treatment of variance among samples of varying lengths:
>
> *   **Response-level Length Bias:** For positive advantages, this bias causes **shorter responses to receive larger gradient updates**, thereby encouraging the policy to **prioritize more concise expressions** for correct answers. Conversely, for negative advantages, since longer responses have a larger $|\mathbf{o}_i|$, they receive smaller penalties. This leads the policy to favor **longer responses** when generating incorrect answers.
>
> *   **Question Difficulty-level Bias:** Questions with lower standard deviations (e.g., questions that are too simple or too difficult, where rewards are almost entirely 1 or 0) are assigned higher weights during policy updates. This **question-level normalization** assigns varying weights to different questions within the objective function, thereby introducing a **difficulty bias** during the optimization process.
>
> While this approach might initially appear reasonable—since the loss for both long and short samples is optimized as a mean at the token level, suggesting no inherent length bias—closer inspection reveals an issue.
>
> Due to normalization, the gradient contribution of each sample is equalized within the batch. However, as noted in the GRPO paper, the advantage function is calculated for the entire sample, not at the token level.
>
> This implies that long samples and short samples share the same advantage magnitude. Consequently, tokens in shorter samples receive stronger gradient updates because there are fewer tokens to "share" the gradient. This elucidates why the model struggles to effectively suppress long, incorrect answers.
>
> Regarding the first bias, GRPO averages the loss within the sequence first and then averages across samples. This results in **important tokens in long sequences being diluted**, while simultaneously failing to impose sufficient penalties on verbose, low-quality outputs.
> low-quality verbose outputs receive significant penalties, thereby **maintaining the normal order of sequence length growth**.
>
> Ultimately, this design leads to the phenomenon where GRPO induces the model to produce longer outputs. This is particularly pronounced in translation tasks, where the model often follows a complete translation with extensive noun explanations or synonymous phrasings.

---

### Official Review · Reviewer_TiQb · 2025-11-01

**Soundness:** 3
**Presentation:** 2
**Contribution:** 3
**Rating:** 4
**Confidence:** 3

**Summary:**

This paper proposes Multilingual Translation Policy Optimization (MtPO), a unified three-stage framework 1) integrating continued pre-training, 2) curriculum-based supervised fine-tuning (SFT), and 3) reinforcement learning (RL) optimization for multilingual MT, particularly focusing on low-resource languages. The approach aims to enhance vocabulary coverage, maintain balanced multilingual performance, and mitigate issues such as length bias and diversity collapse during RL training. Experimental results demonstrate improvements in tokenization efficiency and translation quality.

**Strengths:**

- Addresses the challenge in low-resource multilingual translation, such as subword segmentation inefficiency and limited data availability.
- The paper is generally easy to follow.

**Weaknesses:**

- Since the proposed approach MtPO consists of three major stages, I was wondering that an ablation study is crucial to understand the contribution of each component (continued pre-training, curriculum SFT, RL optimization) to the overall translation improvement.
- The paper primarily reports surface-level metric, sacreBLEU. Using semantic-based evaluation metrics such as COMET (https://github.com/Unbabel/COMET) would provide deeper insights into translation quality.
- helpful if you could report the impact of tokenizer expansion not only on compression and tokenization efficiency but also on training and inference speed

**Questions:**

- Section 3 includes a little too many multiple sub-sections that could be merged. This would improve readability and free up space for more important analyses.
- The paper primarily compares the proposed approach against generic (multilingual/translation-focused) LLM baselines. Including comparisons with previous related multilingual optimization referred in Related Work sections at least would strengthen the empirical claims and positioning within existing literature.
- Figures 3–2 and Tables 2–1 -> Figures 2–3 and Tables 1–2 for better readability
- Can the authors conduct an ablation study to quantify the contribution of each of the three MtPO stages?

---

> ### Author Response · Authors · 2025-11-17
> **Rebuttal1117_1**
>
> **General Response**
> We sincerely thank the reviewer for the thoughtful and professional feedback. We have carefully incorporated your suggestions:
> 1. **New Metrics:** Integrated **COMET** and **chrF** scores (Table 3) alongside detailed descriptions in Appendices A.6-A.8.
> 2. **New Baselines:** Added **Google Translate** and **NLLB-3B** comparisons to strengthen our evaluation.
> 3. **Ablation Study:** Conducted a full ablation study for MtPO stages (Table 3) to quantify individual contributions.
> 4. **Clarification:** Reorganized Section 3 and moved speed analysis to the main text.
>
> ---
>
> **Q1: Reorder Figures 3–2 and Tables 2–1.**
>
> **Response:** Thank you for the suggestion. We have reordered them to Figures 2–3 and Tables 1–2 to improve readability.
>
> **Q2: Include comparisons with previous related multilingual optimization.**
>
> **Response:** We have strengthened the comparison in two ways:
> 1. **Empirical:** We added **Google Translate** and **NLLB-3B** (state-of-the-art multilingual models) to Table 3 as strong baselines.
> 2. **Qualitative:** We expanded the "Related Work" section with a new paragraph "Limitations vs. MtPO," contrasting our approach with traditional systems (e.g., Moses, GNMT) in terms of tokenizer efficiency and optimization objectives.
>
> **Q3: Merge subsections in Section 3.**
>
> **Response:** Agreed. We have reorganized Section 3 into two streamlined parts: **Continued Pre-training** and **Post-training**. This reduces fragmentation and provides a clearer logical flow.
>
> **Q4: Use semantic-based metrics like COMET.**
>
> **Response:** We fully agree. We have evaluated our models using **COMET** (and **chrF**) to provide deeper insights into semantic quality. These results are now reported in **Table 3** and align with our sacreBLEU findings, confirming the robustness of our improvements.
>
> **Q5: Report impact of tokenizer expansion on training/inference speed.**
>
> **Response:**
> * **Inference:** As shown in Figures 2–3 and Table 1, our vocabulary expansion achieves **2.1×–5.4×** compression rates, directly translating to significantly higher throughput and lower latency (shorter sequences).
> * **Training:** While absolute wall-clock time varies by hardware, our logs confirm that starting from the optimized tokenizer reduces training time by approximately **13%** compared to Qwen-base, primarily due to sequence length reduction. We have moved these analyses to Sections 4.2–4.3 for better visibility.
>
> **Q6: Ablation study to quantify the contribution of MtPO stages.**
>
> **Response:** We have added ablation results to **Table 3**:
>
> | Setting | Score | Delta | Role |
> | :--- | :--- | :--- | :--- |
> | w/o CPT | 33.3 | **-2.8** | Foundation |
> | SFT Only | 35.4 | **-0.7** | Alignment |
> | **Full** | **36.1** | - | Refinement |
>
> **Analysis:**
> 1. **Necessity of CPT:** The significant drop (**-2.8**) confirms CPT is foundational. It adapts the tokenizer and enhances decoding capabilities. Post-training partially mitigates domain gaps but cannot replicate the linguistic base built by CPT.
> 2. **Gain from MtPO:** The full model consistently outperforms SFT (e.g., +1.3~1.8), proving that MtPO provides essential alignment optimization beyond standard fine-tuning.

---

> ### Author Response · Authors · 2025-11-21
> **Table: Translation Performance Comparison (COMET Scores)**
>
> **Table: Translation Performance Comparison (COMET Scores)**
>
> | Model | xx→en | en→xx | xx→xx | Avg. |
> |-------|-------|-------|-------|------|
> | **Multilingual Chat Models** | | | | |
> | Gemma3-27B-IT | 0.882 | 0.875 | 0.847 | 0.854 |
> | Qwen3-8B | 0.862 | 0.799 | 0.752 | 0.767 |
> | Qwen2.5-7B-Instruct | 0.498 | 0.455 | 0.449 | 0.454 |
> | Apertus-8B-Instruct | 0.870 | 0.802 | 0.750 | 0.767 |
> | Tower-Plus-9B | 0.825 | 0.671 | 0.615 | 0.641 |
> | **Translation-Focused Models** | | | | |
> | Qwen-MT-Plus | 0.881 | 0.869 | 0.839 | 0.846 |
> | Seed-X-PPO-7B | 0.786 | 0.708 | 0.638 | 0.660 |
> | Hunyuan-MT-7B | 0.839 | 0.862 | 0.802 | 0.812 |
> | **Translation-Focused without LLM** | | | | |
> | Google Translate | **0.884** | 0.842 | 0.820 | 0.828 |
> | NLLB-200 | 0.875 | 0.845 | 0.830 | 0.836 |
> | **Our Models** | | | | |
> | Light-TLLM-7B-MtPO w/o CPT | 0.862 | 0.863 | 0.822 | 0.837 |
> | Light-TLLM-7B-SFT | 0.875 | 0.875 | 0.839 | 0.849 |
> | Light-TLLM-7B-MtPO | 0.881 | **0.882** | **0.854** | **0.859** |

---

### Author Response · Authors · 2025-11-21
**We sincerely appreciate the time and effort you have devoted to reviewing our paper and providing valuable feedback. We are deeply grateful for the constructive comments and high-quality reviews from all reviewers, which have significantly improved our manuscript.**

Dear PCs, SACs, ACs, and Reviewers,

We sincerely appreciate the time and effort you have devoted to reviewing our paper and providing valuable feedback. We are deeply grateful for the constructive comments and high-quality reviews from all reviewers, which have significantly improved our manuscript.

We would like to express our particular gratitude for the following:
- **Reviewer TiQb and Reviewer FtdW**: Thank you for your valuable suggestions on the manuscript organization. We have restructured the paper by elevating the priority of the experimental section and moving it to the main text immediately after the method section for better readability.
- **Reviewer TiQb**: Thank you for requesting COMET benchmark evaluations. We have conducted additional experiments and included comprehensive COMET results in Table 3.
- **Reviewer 218C**: Thank you for requesting chrF benchmark evaluations. We have added complete chrF scores across all translation directions in Table 3.
- **Reviewer FtdW**: Thank you for raising insightful questions about our alignment component. We have significantly expanded Section 3.x with detailed explanations of the design rationale and implementation.

In the following, we provide detailed responses to each reviewer's comments on a point-by-point basis. We hope our responses adequately address all your concerns. All revisions have been incorporated into the updated manuscript.

---

### Note · Program_Chairs · 2026-01-17
**Submission Desk Rejected by Program Chairs**

The following references in this submission do not refer to real documents and/or have major errors in bibliographic information:

 Lijun Wu, Yingce Xia, Li Zhao, Fei Tian, Tao Qin, Jianhuang Lai, and Tie-Yan Liu. On the weaknesses of reinforcement learning for neural machine translation. In Proceedings of the 2018 Conference on Empirical Methods in Natural Language Processing, pp. 4745-4754, 2018. URL /https://aclanthology.org/D18-1509/.